# Intermittent Ca²⁺ signals mediated by Orai1 regulate basal T cell motility

Tobias X Dong[1†], Shivashankar Othy[1†], Milton L Greenberg[1], Amit Jairaman[1], Chijioke Akunwafo[1], Sabrina Leverrier[1], Ying Yu[1], Ian Parker[1,2], Joseph L Dynes[1], Michael D Cahalan[1,3]*

[1]Department of Physiology and Biophysics, University of California, Irvine, Irvine, United States; [2]Department of Neurobiology and Behavior, University of California, Irvine, Irvine, United States; [3]Institute for Immunology, University of California, Irvine, Irvine, United States

**Abstract** Ca²⁺ influx through Orai1 channels is crucial for several T cell functions, but a role in regulating basal cellular motility has not been described. Here, we show that inhibition of Orai1 channel activity increases average cell velocities by reducing the frequency of pauses in human T cells migrating through confined spaces, even in the absence of extrinsic cell contacts or antigen recognition. Utilizing a novel ratiometric genetically encoded cytosolic Ca²⁺ indicator, Salsa6f, which permits real-time monitoring of cytosolic Ca²⁺ along with cell motility, we show that spontaneous pauses during T cell motility in vitro and in vivo coincide with episodes of cytosolic Ca²⁺ signaling. Furthermore, lymph node T cells exhibited two types of spontaneous Ca²⁺ transients: short-duration 'sparkles' and longer duration global signals. Our results demonstrate that spontaneous and self-peptide MHC-dependent activation of Orai1 ensures random walk behavior in T cells to optimize immune surveillance.

DOI: https://doi.org/10.7554/eLife.27827.001

*For correspondence:
mcahalan@uci.edu

†These authors contributed equally to this work

Competing interests: The authors declare that no competing interests exist.

## Introduction

To initiate the adaptive immune response, T cells must make direct contact with antigen-presenting cells (APCs) in the lymph node, enabling T cell receptors (TCRs) to engage peptide-bound MHC molecules presented on the APC surface. Because cognate antigens are rare for any given TCR, many APCs must be scanned to identify those bearing cognate antigens. Thus, optimizing T cell motility to balance search sensitivity, specificity, and speed is crucial for efficient antigen search and proper immune function (*Cahalan and Parker, 2005*; *Krummel et al., 2016*). Both cell-intrinsic and environmental factors have been proposed to regulate T cell motility within lymph nodes and peripheral tissues (*Miller et al., 2002*; *Bousso and Robey, 2003*; *Mempel et al., 2004*; *Mrass et al., 2010*). T cell motility in steady-state lymph nodes under homeostatic conditions, referred to as 'basal motility', has been likened to diffusive Brownian motion, resembling a 'stop-and-go' random walk that results in an overall exploratory spread characterized by a linear mean-squared displacement over time (*Miller et al., 2002*). Subsequent studies defined a role of cellular cues in guiding T cell migration, such as contact with the lymph node stromal cell network or short-term encounters with resident dendritic cells (*Miller et al., 2004*; *Bajénoff et al., 2006*; *Khan et al., 2011*). Whereas the basic signaling mechanisms for cell-intrinsic induction of random motility have been previously explored in fibroblasts and neuroblastoma cells (*Petrie et al., 2009*), it remains unclear if such mechanisms apply in T cells.

Upon T cell recognition of cognate antigen, TCR engagement results in an elevated cytosolic Ca²⁺ concentration that acts as a 'STOP' signal to halt motility and anchor the T cell to the site of antigen presentation (*Donnadieu et al., 1994*; *Negulescu et al., 1996*; *Dustin et al., 1997*;

**eLife digest** To help protect the body from disease, small immune cells called T lymphocytes move rapidly, searching for signs of infection. These signs are antigens – processed pieces of proteins from invading bacteria and viruses – which are displayed on the surface of so-called antigen-presenting cells. To visit as many different antigen-presenting cells as possible, T cells move quickly from one to the next in an apparently random manner. How T cells are programmed to move in this way is largely unknown.

The entry of calcium ions into cells triggers characteristic actions in many cells throughout the body. In T cells, calcium ions enter through Orai1 proteins that form calcium channels on the cell surface. Now, Dong, Othy et al. have asked whether calcium signals guide moving T cells as they search for antigens.

Experiments with individual human T cells in small tubes showed that blocking the Orai1 calcium channels caused the T cells to move faster, because the cells paused less often. The same was seen when human T cells were transplanted into mice.

These findings suggested that calcium signals may indeed guide the T cells' movement, but actually being able to see the calcium signals in the cell would give a much clearer picture of what goes on. To achieve this, Dong, Othy et al. report, in a related study, how they genetically engineered mice to produce a calcium-sensitive reporter protein in their T cells.

Using these new transgenic mice, Dong, Othy et al. could see calcium signals in the T cells before each of the T cell's pauses. Further experiments showed that the calcium signals that control the cell's movements are triggered both by contact with the antigen-presenting cells and internally within the T cells themselves. In another related study, Guichard et al. also conclude that contact with antigen-presenting cells causes calcium signals that control the responses of T cells.

Seemingly random patterns of movement help T cells search for signs of infection, and these new findings reveal a basic part of how T cells are programmed to move in this way. A deeper understanding of T cell movement might allow this process to be controlled. In particular, this knowledge could lead to new treatments for autoimmune diseases, in which T cells incorrectly recognize the body's own antigens as signs of an infection.
DOI: https://doi.org/10.7554/eLife.27827.002

*Bhakta et al., 2005*; *Moreau et al., 2015*). The predominant mechanism for increasing cytosolic $Ca^{2+}$ in T cells is through store-operated $Ca^{2+}$ entry (SOCE), which is mediated by the molecular components STIM1 and Orai1. TCR stimulation triggers depletion of intracellular $Ca^{2+}$ stores in the endoplasmic reticulum (ER), resulting in translocation of the ER-resident $Ca^{2+}$ sensor STIM1 to specialized ER-plasma membrane (PM) junctions where Orai1 channels aggregate into puncta and activate to allow sustained $Ca^{2+}$ influx (*Liou et al., 2005*; *Roos et al., 2005*; *Zhang et al., 2005*; *Luik et al., 2006*; *Vig et al., 2006*; *Zhang et al., 2006*; *Calloway et al., 2009*; *Wu et al., 2014*). Orai1 channel activity is crucial for immune function, as human mutations in Orai1 result in severe combined immunodeficiency (SCID) (*Feske et al., 2006*). Additional roles of Orai1 have been defined in chemotaxis to certain chemokines and T cell homing to lymph nodes (*Greenberg et al., 2013*); actin cytoskeleton rearrangement (*Schaff et al., 2010*; *Dixit et al., 2011*; *Babich and Burkhardt, 2013*; *Hartzell et al., 2016*); migration during shear flow (*Schaff et al., 2010*; *Dixit et al., 2011*); lipid metabolism (*Maus et al., 2017*); and dendritic spine maturation in neurons (*Korkotian et al., 2017*). However, despite their contributions to other aspects of T cell function, no role has been identified for Orai1 channels in T-cell motility patterns underlying scanning behavior.

In this study, we use human and mouse T cells to assess the role of Orai1 and $Ca^{2+}$ ions in regulating basal cell motility. Expression of a dominant-negative Orai1-E106A construct was used to block Orai1 channel activity in human T cells, both in vivo within immunodeficient mouse lymph nodes (*Greenberg et al., 2013*), and in vitro within microfabricated polydimethylsiloxane (PDMS) chambers (*Jacobelli, Friedman et al. 2010*). We use our genetically encoded Salsa6f tandem green/red fluorescent $Ca^{2+}$ indicator (*Dong et al., 2017*) to monitor spontaneous $Ca^{2+}$ signaling in human T cells migrating in confined microchannels in vitro. Finally, using a transgenic mouse strain expressing Salsa6f homozygously in $Cd4^+$ T cells, designated as Cd4-Salsa6f (Hom) mice from here on, we

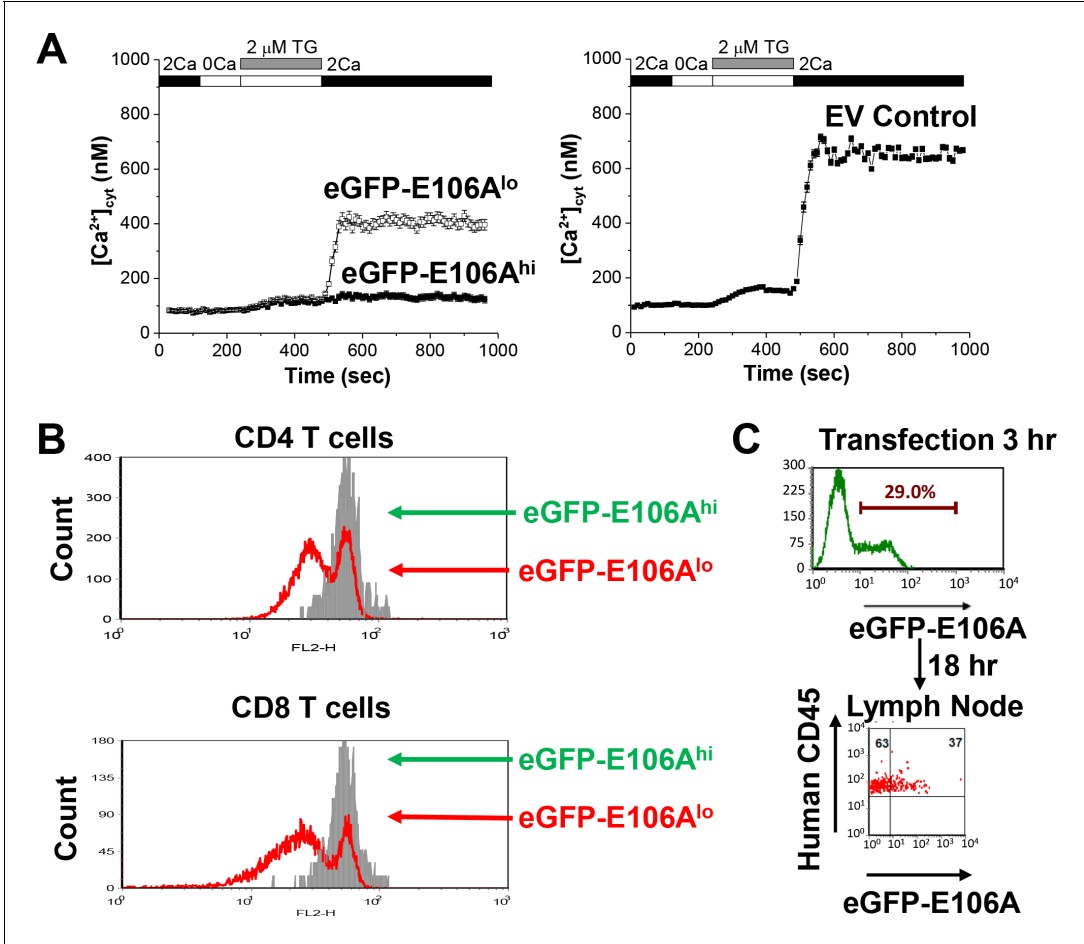

**Figure 1.** Effects of expressing Orai1-E106A on human T cells. (A) Averaged thapsigargin-induced $Ca^{2+}$ entry, measured by fura-2, in activated human CD4[+] T cells transfected with eGFP-Orai1-E106A (left) or empty vector control (EV, right, n = 133 cells); eGFP-E106A transfected cells were grouped into two populations, either eGFP-E106A[hi] with high eGFP fluorescence (solid squares, n = 43 cells) or eGFP-E106A[lo] with no detectable eGFP fluorescence (empty squares, n = 115 cells); bars represent SEM. (B) Primary human CD4[+] and CD8[+] T cells were transfected with eGFP-E106A, then uniformly labeled with the fluorescent cell tracker dye CMTMR and co-cultured with SEB-pulsed primary human dendritic cells from the same donor; proliferation was assessed after 72 hr by CMTMR dilution as measured by flow cytometry. (C) Human CD3[+] T cells were transfected with eGFP-E106A and expression level was measured 3 hr post-transfection before adoptive transfer into reconstituted NOD.SCI.β2 mice; cells were recovered from lymph nodes 18 hr later and eGFP fluorescence was used to measure homing to lymph nodes.

DOI: https://doi.org/10.7554/eLife.27827.003

The following figure supplement is available for figure 1:

**Figure supplement 1.** Protocol for homing and two-photon imaging of transfected human CD3[+] T cells in reconstituted NOD.SCID.β 2 mouse lymph node.

DOI: https://doi.org/10.7554/eLife.27827.004

show that $Ca^{2+}$ signals occur in the absence of specific antigen as T cells crawl in the lymph node. Our results indicate that $Ca^{2+}$ influx, activated intermittently through Orai1 channels, triggers spontaneous pauses during T-cell motility and fine-tunes the random-walk search for cognate antigens.

## Results

### Inhibition of Orai1 in human T cells using a dominant-negative construct

To study the role of Orai1 channel activity in T cell motility, we transfected human T cells with the dominant-negative mutant Orai1-E106A to selectively eliminate ion conduction through the Orai1 pore. The glutamate residue at position 106 in human Orai1 forms the selectivity filter of the Orai1 pore (*Prakriya et al., 2006*; *Vig et al., 2006*; *Yeromin et al., 2006*), and because the Orai1 channel

is a functional hexamer (*Hou et al., 2012*), mutation of E106 to neutrally charged alanine completely inhibits Ca$^{2+}$ permeation in a potent dominant-negative manner (*Greenberg et al., 2013*). Using Fura-2 based Ca$^{2+}$ imaging, we confirmed Orai1 channel block by E106A in activated human T cells transfected with either eGFP-tagged Orai1-E106A or empty vector for control. Thapsigargin-induced SOCE was greatly diminished in cells expressing eGFP-Orai1-E106A, referred to here as eGFP-E106A$^{hi}$ T cells, compared to empty vector-transfected control cells (*Figure 1A*). Ca$^{2+}$ entry was also partially inhibited in a population of transfected T cells with minimal eGFP fluorescence referred to as eGFP-E106A$^{lo}$ cells. To confirm that eGFP-E106A inhibits T cell activation, we challenged transfected human T cells with autologous dendritic cells pulsed with the superantigen Staphylococcal enterotoxin B (*Lioudyno et al., 2008*). T cell proliferation was markedly suppressed in eGFP-E106A$^{hi}$ CD4$^+$ and CD8$^+$ T cells, but not in eGFP-E106A$^{lo}$ T cells (*Figure 1B*). This shows that the residual Orai1 channel activity in eGFP-E106A$^{lo}$ T cells is sufficient for T cell activation and proliferation. Taken together, these experiments show that eGFP-tagged Orai1-E106A expression can serve as a robust tool to assess cellular roles of Orai1 channel activity and that transfected cells without detectable eGFP fluorescence can be used as an internal control.

Orai1 function in human T cell motility was evaluated in vivo using a human xenograft model in which immunodeficient NOD.SCID.β2 mice were reconstituted with human peripheral blood lymphocytes, followed by imaging of excised lymph nodes using two-photon microscopy (*Greenberg et al., 2013*). Reconstitution has been shown to produce a high density of human immune cells within the lymph nodes of immunodeficient mice (*Mosier et al., 1988*), simulating the crowded migratory environment experienced by T cells under normal physiological conditions. Three weeks after reconstitution, human T cells were purified from the same donor, transfected, and adoptively transferred into the reconstituted NOD.SCID.β2 mice (*Figure 1—figure supplement 1*). Whereas control T cells transfected with eGFP showed robust expression and successfully homed to lymph nodes following adoptive transfer 24 hr post-transfection, eGFP-E106A transfected T cells did not home to lymph nodes in the same period, consistent with our previous study indicating that functional Orai1 channel activity is required for T cell homing to lymph nodes (*Greenberg et al., 2013*). To circumvent the homing defect, we injected eGFP-E106A transfected T cells only 3 hr post-transfection, before the expression level of eGFP-E106A had become sufficiently high to block lymph node entry (*Figure 1C*).

## Orai1 block increases human T cell motility within intact lymph node

To evaluate Orai1 function in T cell motility, we imaged human T cells within intact lymph nodes of reconstituted NOD.SCID.β2 mice by two-photon microscopy (*Figure 2A*). We found that eGFP-E106A$^{hi}$ T cells migrated with significantly higher average velocities than co-transferred, mock-transfected CMTMR-labeled T cells (*Figure 2B*). Although both populations had similar maximum and minimum instantaneous cell velocities (*Figure 2C*), eGFP-E106A$^{hi}$ T cells traversed longer distances compared to CMTMR controls (*Figure 2D*), and directionality ratios, a measure of track straightness, decayed more slowly (*Figure 2E*) indicating straighter paths when Orai1 channels were blocked. Orai1-blocked cells displayed shallower turn angles than controls (*Figure 2F*). Furthermore, arrest coefficients, defined by the fraction of time that cell velocity was <2 µm/min, was six-fold lower for eGFP-E106A$^{hi}$ T cells than for control T cells (*Figure 2G*). These differences in motility suggest that the increase in average cell velocity caused by Orai1 block is not due to eGFP-E106A$^{hi}$ T cells moving faster than control T cells, but rather due to a reduced frequency of pausing. Consistent with this interpretation, no eGFP-E106A$^{hi}$ T cells with average velocities <7 µm/min were observed, unlike control T cells in which 23% of average velocities were <7 µm/min (*Figure 2H*).

To replicate our findings in a different immunodeficient mouse model, we repeated our human T cell adoptive transfer protocol using NOD.SCID mice depleted of NK cells. Lymph nodes in these mice are small and contain reticular structures but are completely devoid of lymphocytes (*Shultz et al., 1995*). Similar to experiments on reconstituted NOD.SCID.β2 mice, eGFP-E106A$^{hi}$ human T cells in NOD.SCID lymph nodes migrated with significantly elevated average velocities compared to control T cells (*Figure 2I*), and exhibited lower arrest coefficients (*Figure 2J*). Both eGFP-E106A$^{hi}$ and control T cells migrated at lower speeds in the NK-depleted NOD.SCID model compared to the reconstituted NOD.SCID.β2 model. Because control human T cells in reconstituted NOD.SCID.β2 lymph nodes migrated at similar speeds to wild-type mouse T cells in vivo (*Miller et al., 2002*), reconstitution results in a lymph node environment that more closely mimics

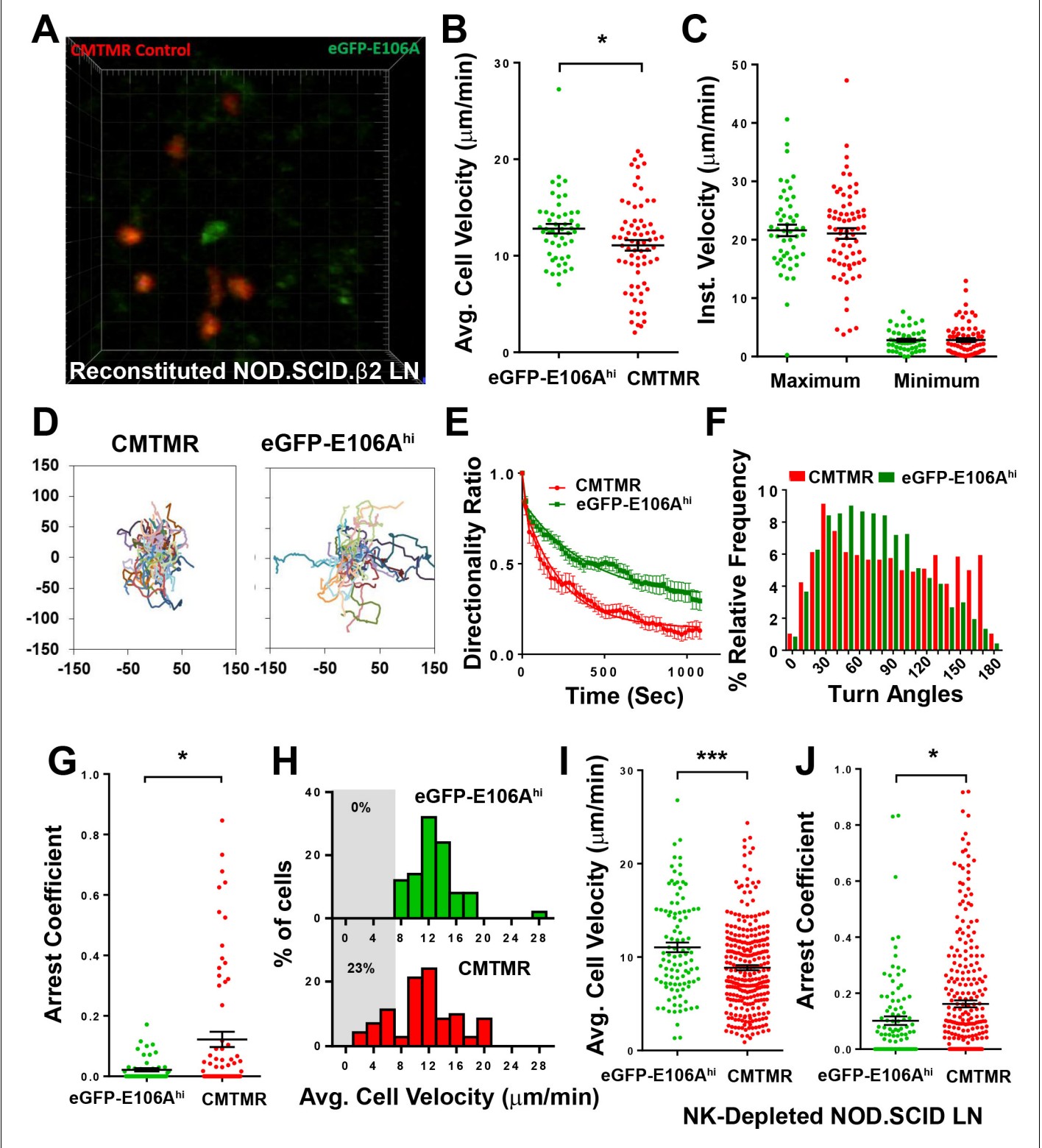

**Figure 2.** Orai1 block increases human T cell motility within reconstituted NOD.SCID.β2 lymph nodes. (**A**) Two-photon microscopy of migrating human T cells, showing eGFP-E106A transfected cells in green and CMTMR-labeled mock transfected cells in red, within intact mouse lymph node 18 hr after adoptive co-transfer of $5 \times 10^6$ of each cell type. (**B**) Average cell velocities of eGFP-E106A$^{hi}$ (n = 50) versus CMTMR-labeled control (n = 71) T cells; bars represent mean ± SEM, data from independent experiments using four different donors (12.8 ± 0.5 μm/min vs. 11.1 ± 0.5 μm/min for E106A$^{hi}$ vs CMTMR cells, p=0.0268). (**C**) Maximum and minimum cellular instantaneous velocities of eGFP-E106A$^{hi}$ (green) versus CMTMR-labeled (red) control T

*Figure 2 continued on next page*

*Figure 2 continued*

cells. (Hodges-Lehmann median difference of −0.21 µm/min, −2.82 to 2.16 µm/min 95% confidence interval for maximum velocity and −0.36 µm/min, −1.02 to 0.36 µm/min 95% confidence interval for minimum velocity) (D) Superimposed tracks with their origins normalized to the starting point. Cells were tracked for more than 20 min. n = 111 (CMTMR), n = 58 (eGFP-E106A[hi]) (E) Directionality ratio (displacement/distance) over elapsed time. For Orai1-blocked cells in green, tau = 397 s; vs CMTMR controls in red, tau = 238 s, n = 49 time points. (F) Histogram of turn angles in eGFP-E106A[hi] (green) and CMTMR controls (red). Mean ± SEM, 74.5 ± 1.0 degrees for Orai1 blocked cells vs 86.5 ± 1.5 degrees for CMTMR controls, p=0.0001, two-tailed T test. (G) Arrest coefficients of eGFP-E106A[hi] compared with CMTMR-labeled control T cells, defined as fraction of time with instantaneous velocity <2 µm/min. (For Orai1-blocked cells in green, 0.02 ± 0.01; vs. CMTMR controls in red, 0.12 ± 0.03, p=0.0406) (H) Frequency distribution of average cell velocities for eGFP-E106A[hi] (top) and CMTMR-labeled control T cells (bottom), cells with average velocity <7 µm/min are highlighted in gray; tick marks denote the center of every other bin. (I,J) Average cell velocities (I) and arrest coefficients (J) of eGFP-E106A[hi] (green, n = 102) vs CMTMR-labeled control (red, n = 278) human T cells in NK-cell-depleted immunodeficient mouse lymph nodes. Average cell velocities: 11.0 ± 0.5 µm/min vs. 8.8 ± 0.3 µm/min, p=0.0004; Arrest coefficients: 0.10 ± 0.02 vs. 0.16 ± 0.01, p=0.0516 for E106A[hi] vs CMTMT cells; bars represent mean ± SEM, data from independent experiments using eight different donors, ***p<0.005.

DOI: https://doi.org/10.7554/eLife.27827.005

normal physiological conditions. Furthermore, the greater effect of Orai1 block on T cell arrest coefficients in crowded reconstituted lymph nodes suggests that Orai1's role in motility is more pronounced in crowded cell environments.

## Orai1 channel activity triggers pauses during human T cell motility in vitro in the absence of extrinsic cell contact

To evaluate whether the pronounced effect of Orai1 channel block on the arrest coefficient in reconstituted lymph nodes was a result of environmental factors such as increased cellular contacts or increased confinement, we tracked human T cells in microfabricated PDMS chambers with cell-sized microchannels 7 µm high x 8 µm wide. These ICAM-1 coated microchannels simulate the confined environment of densely packed lymph nodes (*Jacobelli et al., 2010*), while eliminating possible cell-extrinsic factors. Transfected human T cells were activated with plate-bound anti-CD3/28 antibodies and soluble IL-2, then dropped into chambers and monitored by time-lapse confocal microscopy, using phase contrast to visualize eGFP-E106A[lo] T cells (*Figure 3A,B*). Upon entry into microchannels, eGFP-E106A[hi] T cells migrated with higher average cell velocities than eGFP-E106A[lo] T cells (*Figure 3C*), similar to our in vivo findings from intact lymph node (*Figure 2*). To ensure that the observed difference in cell velocity was due to suppressed Orai1 channel function and not overexpression of Orai1 protein, we also tracked T cells transfected with eGFP-tagged wild-type Orai1. Both eGFP-Orai1[hi] and eGFP-Orai1[lo] T cells migrated at the same average cell velocity (*Figure 3C*), demonstrating that Orai1 channel overexpression, in itself, does not perturb T cell motility in microchannels. Since eGFP-E106A[lo] T cells have reduced Orai1 channel activity but still retain the same cell velocity as eGFP-Orai1 transfected T cells (c.f., *Figures 1A* and *3C*), this suggests that partial Orai1 function is sufficient to generate normal pausing frequency in confined environments. The frequency distribution of cell velocities in vitro is comparable to our in vivo data: fewer GFP-E106A[hi] T cells migrated with average cell velocities <7 µm/min as compared to eGFP-E106A[lo] T cells (11% vs 29%; *Figure 3D*). Furthermore, eGFP-E106A[hi] T cells exhibited lower arrest coefficients (*Figure 3E*) and less variation in velocity than eGFP-E106A[lo] T cells (*Figure 3F*). Although eGFP-E106A[hi] T cells had lower arrest coefficients, the durations of their pauses were not significantly different than in eGFP-E106A[lo] T cells (*Figure 3G*). Taken together, the reduced arrest coefficients in eGFP-E106A[hi] T cells indicate that inhibition of Orai1 channel activity results in reduced frequency of pauses during T cell motility. These in vitro results confirm our in vivo findings and support the hypothesis that Orai1 activity intermittently triggers cell arrest, resulting in an overall decrease in motility within confined environments. Moreover, since our in vitro microchannel assay eliminates extrinsic cell-cell interactions, this indicates that Orai1 can be spontaneously activated to modulate T cell motility.

## Spontaneous Ca$^{2+}$ signals during confined motility in vitro are correlated with reduced T cell velocity

To study the correlation between Ca$^{2+}$ signals and T cell motility, human CD4$^+$ T cells were transfected with Salsa6f, a novel genetically encoded Ca$^{2+}$ indicator consisting of tdTomato fused to GCaMP6f, activated the T cells for 2 days with plate-bound anti-CD3/28 antibodies, then dropped

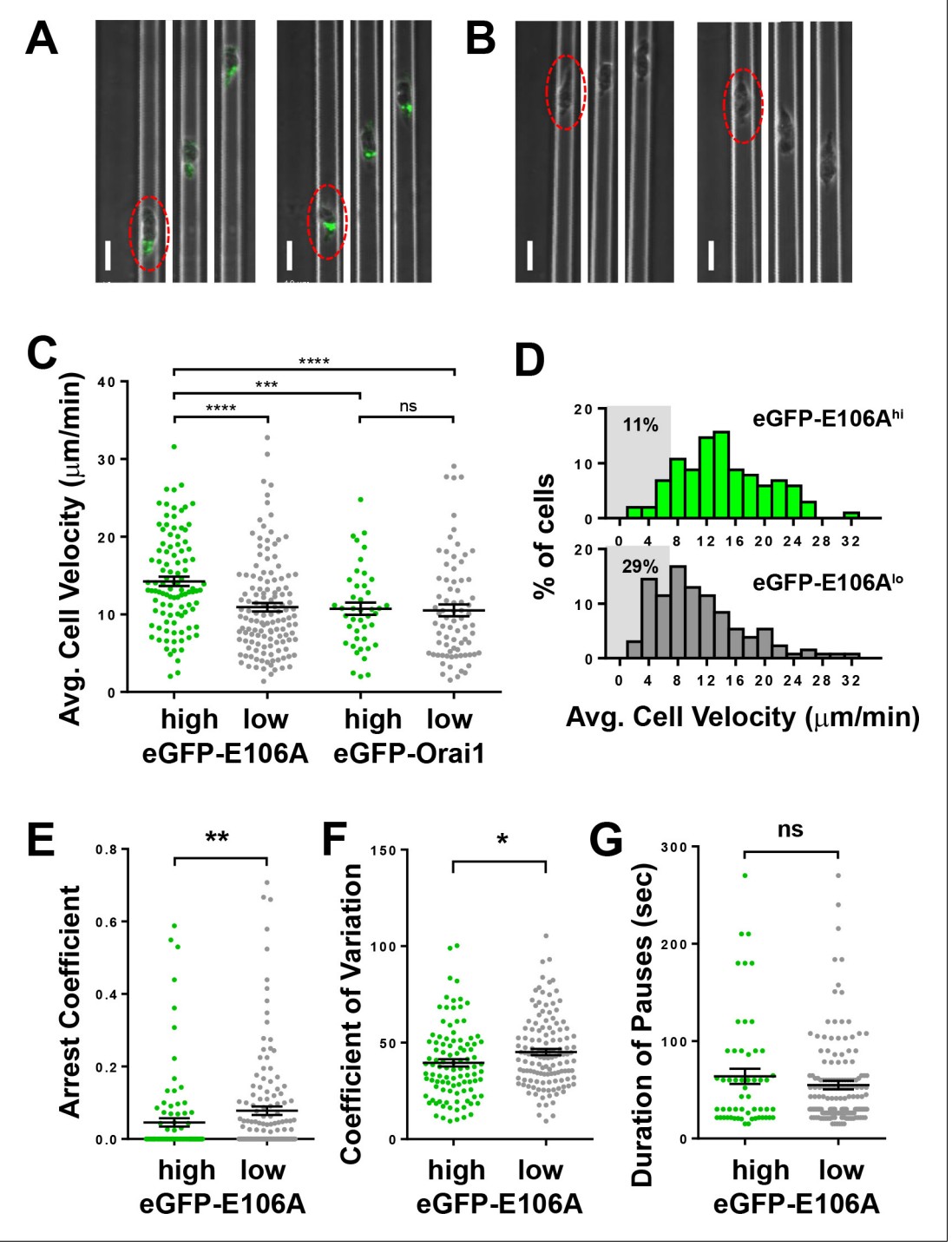

**Figure 3.** Orai1 block reduces frequency of pausing during human T cell motility in vitro. (**A,B**) Confocal microscopy of eGFP-E106A transfected human CD4$^+$ T cells in microfabricated channels 7 μm high by 8 μm wide, showing two individual eGFP-E106A$^{hi}$ T cells (**A**) and two eGFP-E106A$^{lo}$ T cells (**B**), each circled in red in the first frame; individual images taken 1 min apart, scale bar = 10 μm. (**C**) Comparison of average cell velocities of eGFP-E106A transfected T cells (eGFP-E106A$^{hi}$ cells in green, n = 102; eGFP-E106A$^{lo}$ cells in gray, n = 131; 14.2 ± 0.6 μm/min vs. 10.9 ± 0.5 μm/min, p<0.0001 for E106A$^{hi}$ vs E106A$^{lo}$ cells) and eGFP-Orai1 transfected control T cells (eGFP-Orai1$^{hi}$ cells in green, n = 43; eGFP-Orai1$^{lo}$ cells in gray, n = 76; 10.7 ± 0.8 μm/min vs. 10.5 ± 0.8 μm/min for Orai1$^{hi}$ vs Orai1$^{lo}$ cells; Hodges-Lehmann median difference of −0.84 μm/min, −2.96 to 1.28 μm/min 95% CI); bars represent mean ± SEM, data from independent experiments using five different donors. (**D**) Frequency distribution of average cell velocities of eGFP-E106A$^{hi}$ (top) and eGFP-E106A$^{lo}$ (bottom) human T cells, cells with average

*Figure 3 continued on next page*

*Figure 3 continued*

velocity <7 µm/min are highlighted in gray; tick marks denote the center of every other bin. (E) Arrest coefficients of eGFP-E106A[hi] vs eGFP-E106A[lo] human T cells, defined as fraction of time each individual cell had an instantaneous velocity <2 µm/min (0.05 ± 0.01 vs. 0.08 ± 0.01 for E106A[hi] vs E106A[lo] cells, p=0.0015). (F) Variance in velocity of eGFP-E106A[hi] vs eGFP-E106A[lo] human T cells, coefficient of variation is calculated by standard deviation divided by the mean of instantaneous velocity for each individual cell (39.5 ± 1.9% vs. 45.1 ± 1.6% for E106A[hi] vs E106A[lo] cells, p=0.0138). (G) Duration of pauses for eGFP-E106A[hi] vs eGFP-E106A[lo] human T cells (Hodges-Lehmann median difference of 0 s, −8.43 to 4.71 s 95% CI for E106A[hi] vs E106A[lo] cells); bars represent mean ± SEM, *p<0.05, **p<0.01, ***p<0.005, ****p<0.001.

DOI: https://doi.org/10.7554/eLife.27827.006

into ICAM-1 coated microchambers. As previously shown (*Dong et al., 2017*), Salsa6f is localized to the cytosol, with red fluorescence from tdTomato that reflects fluctuations in cell movement and very low baseline green fluorescence from GCaMP6f that rises sharply during $Ca^{2+}$ signals (*Figure 4*). Salsa6f-transfected human T cells were tracked in both confined microchannels (*Figure 4A*, *Video 1*) and the open space adjacent to entry into microchannels (*Figure 4C*, *Video 2*), to evaluate T cell motility under varying degrees of confinement. Intracellular $Ca^{2+}$ levels were monitored simultaneously using the ratio of total GCaMP6f fluorescence intensity over total tdTomato fluorescence intensity (designated as G/R ratio), enabling detection of a notably stable baseline ratio unaffected by motility artifacts in moving T cells while reporting spontaneous $Ca^{2+}$ signals that could be compared to changes in motility (*Figure 4B,D*, orange and black traces, respectively).

Human T cells expressing Salsa6f migrating in confined microchannels exhibited sporadic $Ca^{2+}$ signals as brief peaks unrelated to changes in cell velocity, or as more sustained periods of $Ca^{2+}$ elevation associated with reduced cell velocity (*Figure 5A,B*). To evaluate the correlation between T cell velocity and $Ca^{2+}$ signals, we compared average T cell velocities during periods of sustained $Ca^{2+}$ elevations to average velocities at baseline $Ca^{2+}$ levels. T cell velocity decreased significantly when cytosolic $Ca^{2+}$ was elevated above baseline (5.9 ± 0.1 µm/min vs. 10.0 ± 0.1 µm/min, p<0.0001; *Figure 5C*). $Ca^{2+}$ signaling episodes that last for 30 s or longer accompany and appear to closely track the duration of pauses in cell movement. Comparison of instantaneous velocities with corresponding cytosolic $Ca^{2+}$ signals (G/R ratio) by scatter plot revealed a strong inverse relationship: highly motile T cells always exhibited baseline $Ca^{2+}$ levels, while elevated $Ca^{2+}$ levels were only found in slower or arrested T cells (*Figure 5D*). It is important to note that these $Ca^{2+}$ signals and reductions in velocity occurred in the absence of any extrinsic cell contact or antigen recognition, indicating that $Ca^{2+}$ elevations, like pausing and Orai1 activation, can be triggered in a cell-intrinsic manner.

To compare the effects of Orai1 activity on the motility of T cells in a less confined environment, we also monitored T cell migration within the

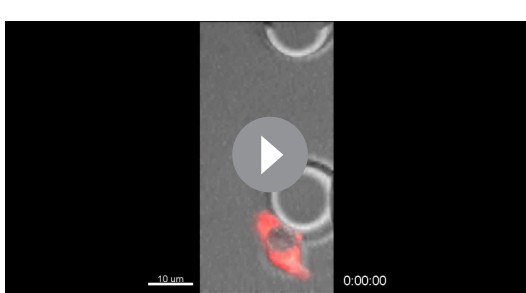

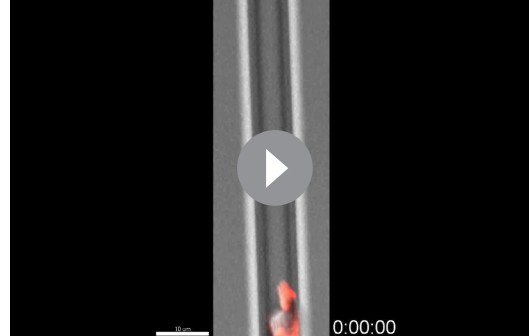

**Video 2.** Salsa6f transfected human T cells in open microchamber, with merged red (tdTomato), green (GCaMP6f), and DIC channels, circular structures are support pillars part of the PDMS microchamber; scale bar = 10 µm, time shown in hr:min:s. This video corresponds to *Figure 4C*.

DOI: https://doi.org/10.7554/eLife.27827.009

**Video 1.** Salsa6f-transfected human T cell in confined microchannel. Merged red (tdTomato), green (GCaMP6f), and DIC channels; scale bar = 10 µm, time shown in hr:min:s. This video corresponds to *Figure 4A*.

DOI: https://doi.org/10.7554/eLife.27827.008

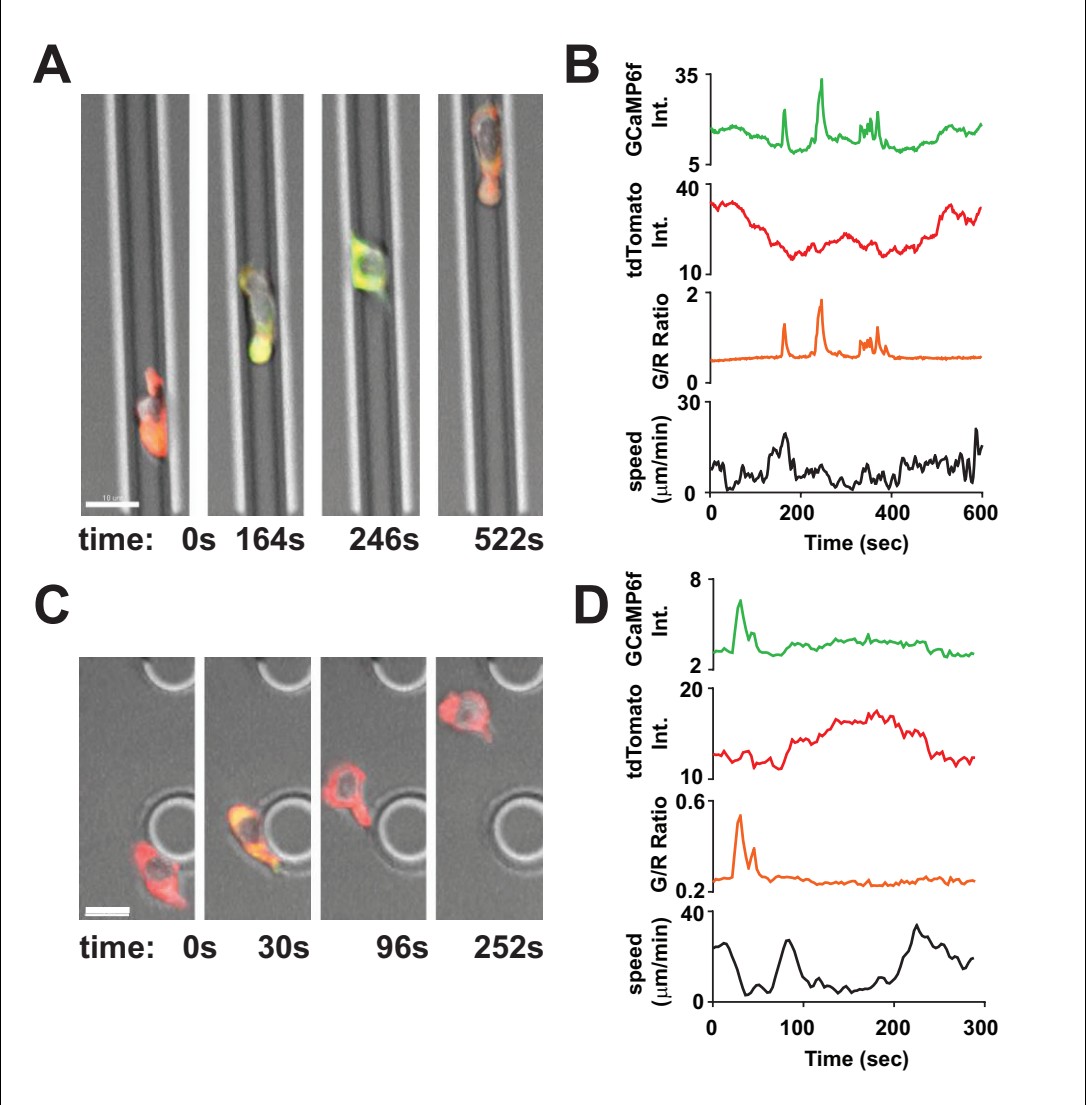

**Figure 4.** Tracking Ca$^{2+}$signals in human T cells in vitro with Salsa6f. (**A,C**) Confocal microscopy of Salsa6f transfected human CD4$^+$ T cells in ICAM-1 coated microchannels 7 μm high by 8 μm wide (A, *Video 1*) and open space (C, *Video 2*), showing merged red (tdTomato), green (GCaMP6f), and DIC channels; circular structures shown in (**C**) are support pillars part of the PDMS chamber; scale bar = 10 μm, time = s. (**B,D**) Total intensity tracings of GCaMP6f (green) and tdTomato (red) fluorescence, G/R ratio (orange), and speed (black), for corresponding T cells shown in (**A**) and (**C**); data representative of independent experiments from three different donors.

DOI: https://doi.org/10.7554/eLife.27827.007

open space in PDMS chambers adjacent to entry into microchannels (*c.f.*, *Figure 4A,C*). We reasoned that in this two-dimensional space with reduced confinement, T cells may not gain sufficient traction for rapid motility, and instead may favor integrin-dependent sliding due to increased exposure to the ICAM-1-coated surface (*Krummel et al., 2014*). In addition, the same population of T cells could be tracked as they migrated into and along the confined microchannels, providing a valuable internal control. We found that eGFP-E106A$^{hi}$ T cells migrated with similar velocities to eGFP-E106A$^{lo}$ T cells in the open space, but these eGFP-E106A$^{hi}$ T cells still exhibited higher motility in the microchannels than eGFP-E106A$^{lo}$ T cells (*Figure 5E*). Furthermore, Salsa6f-transfected T cells within the open space rarely produced Ca$^{2+}$ transients (*c.f.*, *Figure 5D,F*, top left quadrants, 13% of the time in microchannels vs 2% in open space), implying that Ca$^{2+}$ elevations, and by extension, Orai1 channel activity, do not generate pauses when T cells are reliant on integrin binding for

motility. Consistent with this, differentiated Th1 cells from Cd4-Salsa6f mice also showed similar instantaneous velocities and only rare $Ca^{2+}$ transients when plated on open-field ICAM-coated coverslips (*Figure 5—figure supplement 1*). Taken together, these experiments establish a role for Orai1 channels and $Ca^{2+}$ influx in modulating T cell motility within confined environments.

## Spontaneous T cell $Ca^{2+}$ transients during basal motility in the lymph node

Using Salsa6f, expressed in a *Cd4^Cre*-dependent transgenic model we have reported that mouse T cells exhibit frequent transient $Ca^{2+}$ signals ('sparkles') in homeostatic lymph nodes in the absence of specific antigen (*Dong et al., 2017*). To further analyze the relationship between $Ca^{2+}$ signaling and motility in detail within lymph nodes, we adoptively transferred homozygotic Cd4-Salsa6f (Hom) T cells into congenic mice and, using two-photon microscopy in explanted recipient lymph nodes, tracked the red tdTomato signal to establish cell position and the green CGaMP6f signal as a measure of cytosolic $Ca^{2+}$. First, to delineate any adverse effect of Salsa6f on homing and in situ motility of T lymphocytes, we co-injected equal numbers of Cd4-Salsa6f and *Cd4^Cre* control cells into WT recipients (*Figure 6A*). For simultaneous imaging and to normalize any dye toxicity, Cd4-Salsa6f and *Cd4^Cre* T cells were labeled with CellTrace Yellow (CTY) and CellTrace Violet (CTV), respectively. Comparable numbers of input cells were recovered from the subcutaneous lymph nodes after 18 hr (*Figure 6B*). Two-photon imaging and tracking in lymph nodes showed typical stop and go motility and meandering cell tracks (*Figure 6C,D*, *Video 3*) for both cell types. Instantaneous 3D velocities (*Figure 6E*) and mean track velocities (*Figure 6F*) were indistinguishable, as was the decay rate of directionality ratio *(Figure 6G)*. Furthermore, mean-squared displacement (MSD) time analysis showed random-walk behavior for both cell types with similar motility coefficients (*Figure 6H,I*). Altogether, motility characteristics of Salsa6f T cells are indistinguishable from control T cells.

To determine whether spontaneously occurring $Ca^{2+}$ signals are correlated with motility, we transferred Cd4-Salsa6f cells alone into wild-type recipients and tracked red and green fluorescence intensities in the lymph nodes after 18 hr.

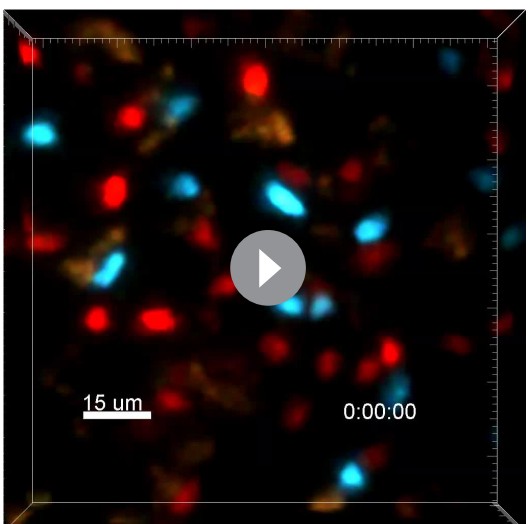

**Video 3.** Motility of Cd4-Salsa6f T cells in lymph node following adoptive transfer. *Cd4^Cre* and Cd4-Salsa6f cells and their trails are shown in teal and in red, respectively. Autofluorescent bodies appear as faint stationary yellow structures. Images were acquired at ~11 s interval. Playback speed = 50 frames per second; time shown in hr:min:sec. Video corresponds to *Figure 6C*.

DOI: https://doi.org/10.7554/eLife.27827.013

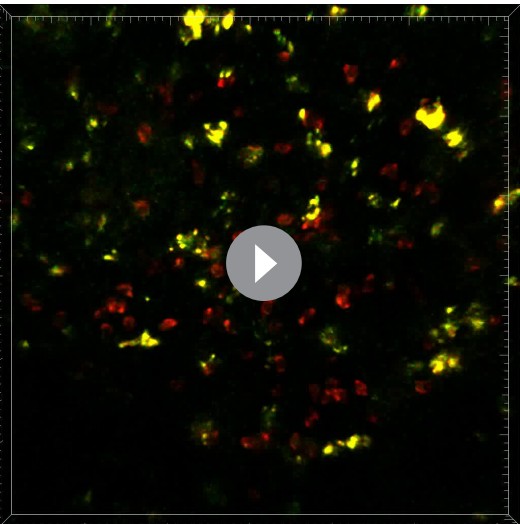

**Video 4.** Calcium signals in adoptively transferred Cd4-Salsa6f T cells. Red signal from tdTomato expression in cytosol facilitates identification and tracking of cells; green GCaMP6f signal detects elevation of $Ca^{2+}$. Autofluorescent structures appear as stationary yellow bodies. Movie is paused at frame 323, zoomed in to emphasize two examples of $Ca^{2+}$ transients and an autofluorescent body. Images were acquired at 5 s interval. Major tick marks at 20 μm. Playback speed = 50 frames/s, time shown in hr:min:s. Video corresponds to *Figure 7A*.

DOI: https://doi.org/10.7554/eLife.27827.015

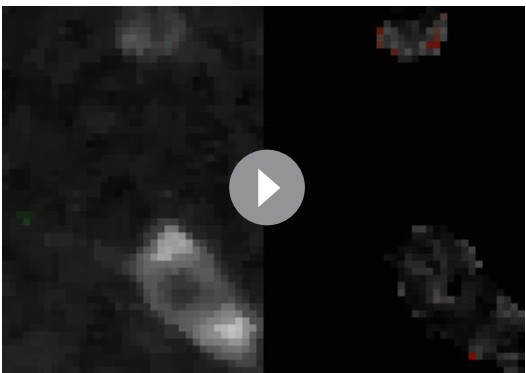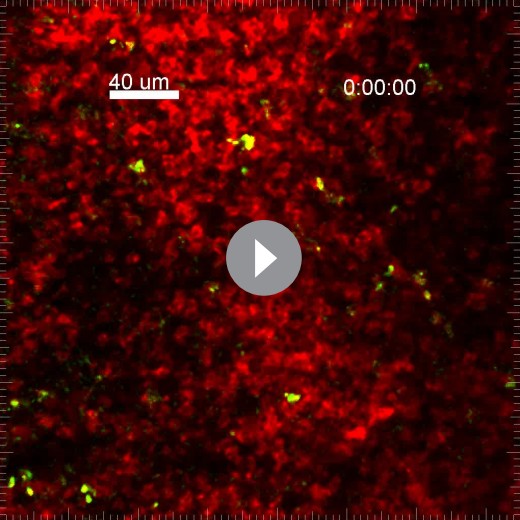

**Video 5.** A brief Ca²⁺signal filling the back of a moving adoptively transferred Salsa6f (Hom) T cell. Left: composite of red tdTomato fluorescence pseudocolored grayscale with green GCaMP6f fluorescence. Right: Corresponding Green/Red ratios, masked to red channel as in *Figure 7C*. Images acquired at 1 frame every 5 s and 0.5 microns/pixel. Playback speed = 3 frames per second. 0.5 μm/pixel. Video corresponds to *Figure 7C*.

DOI: https://doi.org/10.7554/eLife.27827.016

**Video 6.** Calcium transients in steady state lymph nodes. Cd4-Salsa6f (Hom) lymph node imaged at 0.5 s interval, processed to visualize Ca²⁺ transients (sparkles and cell-wide) in green. Red channel is turned off after beginning to facilitate viewing of Ca²⁺ transients. Autofluorescent structures appear as stationary green bodies. Playback speed = 100 frames/s. time shown in hr:min:s. Video corresponds to *Figure 8A*

DOI: https://doi.org/10.7554/eLife.27827.018

Consistent with our previous observation, adoptively transferred T cells retained Salsa6f indicator in their cytosol, and Ca²⁺ signals were readily observed in motile Salsa6f⁺ T cells (*Figure 7A*, *Video 4*). We monitored the G/R ratios over time and observed a strong negative correlation between instantaneous cell velocity and Ca²⁺ levels (*Figure 7B*). By examination of fluctuating cell velocity traces with corresponding G/R ratios, we found that the Ca²⁺ rise is clearly associated with a decrease in velocity (*Figure 7C and D*, *Video 5*). Notably, on average, peaks of Ca²⁺ transients precede the average cell velocity minimum, suggesting that spontaneous rise in intracellular Ca²⁺ levels leads to cell pausing (*Figure 7E*).

## Frequency, duration and MHC dependence of T cell Ca²⁺ transients in homeostatic lymph nodes

Imaging adoptively transferred T cells in recipient lymph nodes is an ideal approach to probe in vivo T-cell motility. However, this approach is limiting when it comes to identifying the abundance and duration of Ca²⁺ signaling events, because transferred cells label only a fraction of the lymph node (<1%) and longer imaging intervals are required to collect sufficient volume of 4D data (>5 s). Therefore, to measure the endogenous frequency and duration Ca²⁺ transients, we imaged homeostatic Cd4-Salsa6f (Hom) lymph nodes at two frames per second. All endogenous T cells (Cd4⁺ and Cd8⁺) are labeled with the Salsa6f probe in Cd4-Salsa6f lymph nodes because T cells go through the double-positive stage during development in the thymus. More than 800 Ca²⁺ transients were identified in a 300 × 300 μm area in a 10-min interval. We identified two types of Ca²⁺ transients: numerous small and brief spots (sparkles); and less frequent large, cell-wide transients (*Figure 8A*, *Video 6*). Consistent with our previous report (*Dong et al., 2017*), most Ca²⁺ transients were localized to small regions of the cell and of short duration, spanning 2 μm² in area (*Figure 8B*) and lasting about 2 s (*Figure 8C*). Altogether, the strong association of Ca²⁺ transients with reductions in cell velocity leading to pausing, and the sheer number of Ca²⁺ transients in homeostatic lymph nodes suggest that cytosolic Ca²⁺ is a key regulator of basal cellular motility under steady-state conditions in the absence of specific antigen.

Lymphocytes migrate in the immune dense micro-environment of secondary lymphoid tissues, constantly interacting with other immune cells, including resident antigen presenting cells (*Germain et al., 2012*). Indeed, constant recognition of low levels of self-antigens through T cells

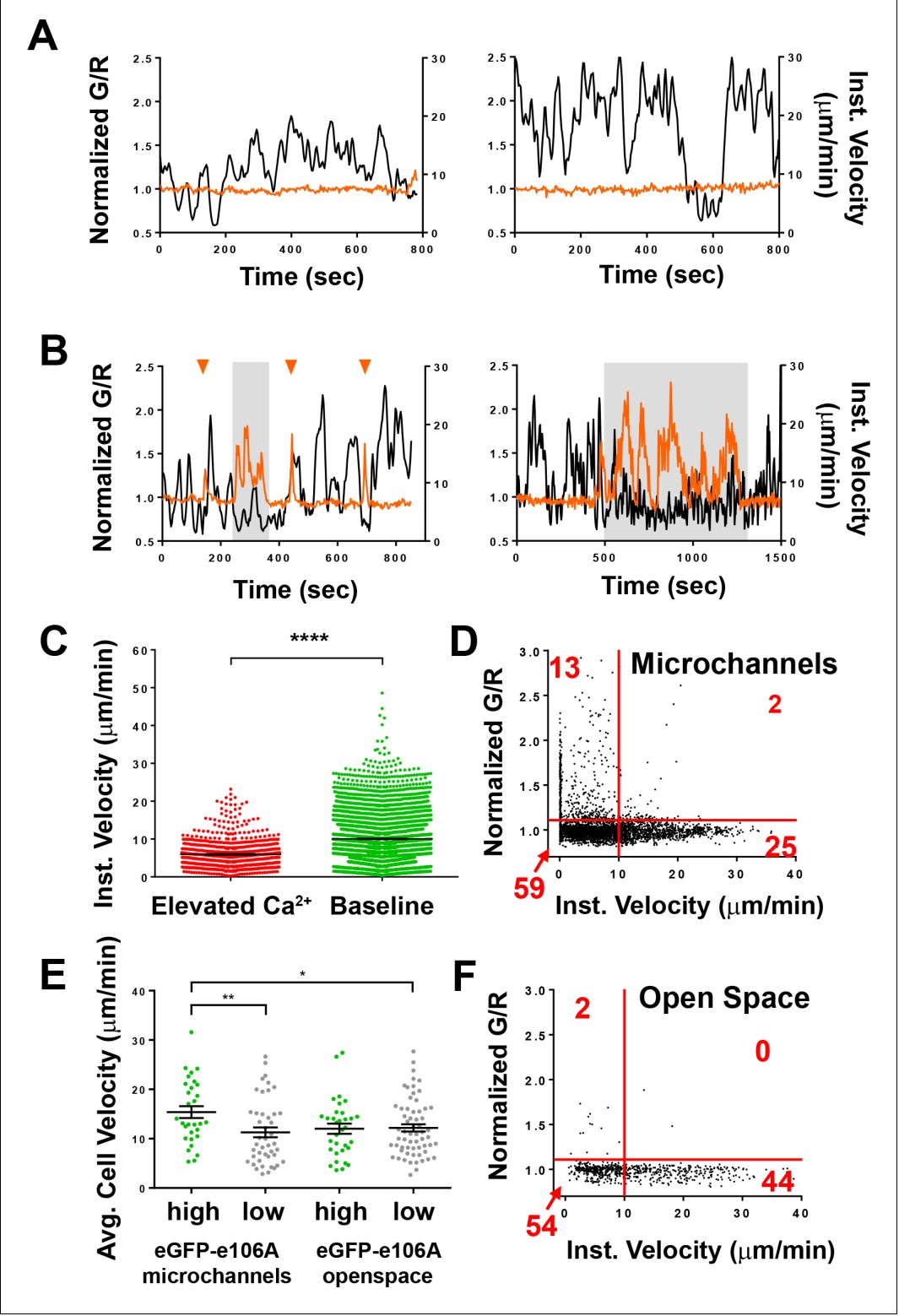

**Figure 5.** Spontaneous Ca$^{2+}$signals during human T cell motility in vitro are correlated with reduced velocity. (A, B) Sample tracks from Salsa6f-transfected human T cells in microchannels, with intracellular Ca$^{2+}$ levels as G/R ratios for each time point, normalized to zero-time (orange), overlaid with instantaneous cell velocity (black), cells in (A) have stable Ca$^{2+}$ levels, cells in (B) show brief Ca$^{2+}$ transients (arrowheads) or sustained Ca$^{2+}$ signaling (gray highlights). (C) Instantaneous velocity of Salsa6f-transfected human T cells in microchannels during elevated

*Figure 5 continued on next page*

*Figure 5 continued*

cytosolic Ca$^{2+}$ levels (red) and during basal Ca$^{2+}$ levels (green); n = 22 cells, data from independent experiments using three different donors; ****p<0.001. (D) Scatter plot of Salsa transfected human T cells in microchannels, instantaneous cell velocity versus normalized G/R ratio for each individual time point analyzed; red numbers in each quadrant show percent of time points, split by 1.10 normalized G/R ratio and 10 µm/min; n = 4081 points. (E) Mean track velocity of eGFP-E106A-transfected human T cells, comparing eGFP-E106A$^{hi}$ (green) and eGFP-E106A$^{lo}$ T cells (gray) in confined microchannels vs open space; n = 30, 44, 33, and 62 cells, respectively (15.4 ± 1.2 µm/min vs. 11.3 ± 1.0 µm/min for E106A$^{hi}$ vs E106A$^{lo}$ cells in microchannels; p=0.0099 and 12.0 ± 1.0 µm/min vs. 12.2 ± 0.7 µm/min for E106A$^{hi}$ vs E106A$^{lo}$ cells in open space; Hodges-Lehmann median difference of 0.15 µm/min, −2.46 to 2.40 µm/min 95% CI). Bars represent mean ± SEM, data from independent experiments using two different donors, *p<0.05, **p<0.01. (F) Scatter plot of Salsa transfected human T cells in open space, instantaneous cell velocity versus GCaMP6f/tdTomato R/R$_0$ for each individual time point analyzed; red numbers in each quadrant show percent of cells, split by 1.10 normalized G/R ratio and 10 µm/min; n = 723 points.
DOI: https://doi.org/10.7554/eLife.27827.010

The following figure supplement is available for figure 5:

**Figure supplement 1.** Tracking cell motility and Ca$^{2+}$signals in Cd4-Salsa6f T cells on ICAM coated coverslips.
DOI: https://doi.org/10.7554/eLife.27827.011

receptor (TCR)-pMHC interactions is critical for maintaining sensitivity to foreign antigens (*Stefanová et al., 2002*); and deprivation (>7 days) of pMHC-II signals impairs T cell motility (*Fischer et al., 2007*). To investigate whether Ca$^{2+}$ signals in steady state lymph nodes are result of self-peptide recognition, we blocked MHC Class I and II signaling for 48 hr in Cd4-Salsa6f (Hom) lymph nodes. The number of cell-wide events was not significantly different (p=0.06), whereas the sparkle frequency was significantly decreased (p=0.02) in MHC-blocked lymph nodes compared to isotype control (ITC) antibody treatment (*Figure 9A,D–G*). There was also significant variation in the number of Ca$^{2+}$ transients in ITC antibody and uninjected controls (Coefficient of variation = 41% to 45%), which may be due to the presence of heterogeneous antigen presenting cells displaying varying amount of self-peptides during steady-state. Most notably, however, a significant number of Ca$^{2+}$ transients remained even after MHC block, which we believe reflects a basal level of spontaneous Ca$^{2+}$ activity independent of antigen recognition. In contrast, the intensity of individual Ca$^{2+}$ transients in MHC blocked lymph nodes did not differ significantly from the ITC controls (*Figure 9B,C*). Altogether, our data indicate that T cells display substantial spontaneous Ca$^{2+}$ transients even in absence of self-peptide recognition, suggesting a role in regulating basal T lymphocyte motility.

## Discussion

In this study, we demonstrate that Orai1 channel activity regulates motility patterns that underlie immune surveillance. Human T cells expressing the dominant-negative Orai1-E106A construct migrated with higher average velocities than controls, both in reconstituted mouse lymph nodes in vivo and in confined microchannels in vitro. In particular, we found that the increase in average cell velocity was not due to an increase in maximum cell velocity, but to a reduced frequency of cell pausing accompanied by increased directional persistence, resulting in straighter paths. Human T cells demonstrate Ca$^{2+}$ transient-associated and Orai1-dependent pauses in vitro within confined microchannels devoid of cell-extrinsic factors. Furthermore, we use a novel ratiometric genetically encoded Ca$^{2+}$ indicator, Salsa6f, along with T cells from Cd4-Salsa6f(Hom) transgenic mice, to show that intermittent Ca$^{2+}$ signals coincide with reduced cell velocity. Treatment of Cd4-Salsa6f (Hom) mice with MHC class-I and -II blocking antibodies substantially reduces but does not eliminate the frequent T cell Ca$^{2+}$ transients seen in lymph nodes. Based on these findings, we propose that Orai1 channel activity regulates the timing of stop-and-go motility in T cells and tunes the search for cognate antigen in the crowded lymph node.

Our Orai1-E106A construct derives its specificity and potency by targeting the pore residue responsible for the channel selectivity filter. Incorporation of Orai1-E106A likely blocks heterodimers of Orai1 and other channels, such as Orai2 or Orai3, in addition to homomeric Orai1 channels. Very recent evidence demonstrates the existence of heteromeric channels composed of Orai1 and Orai2

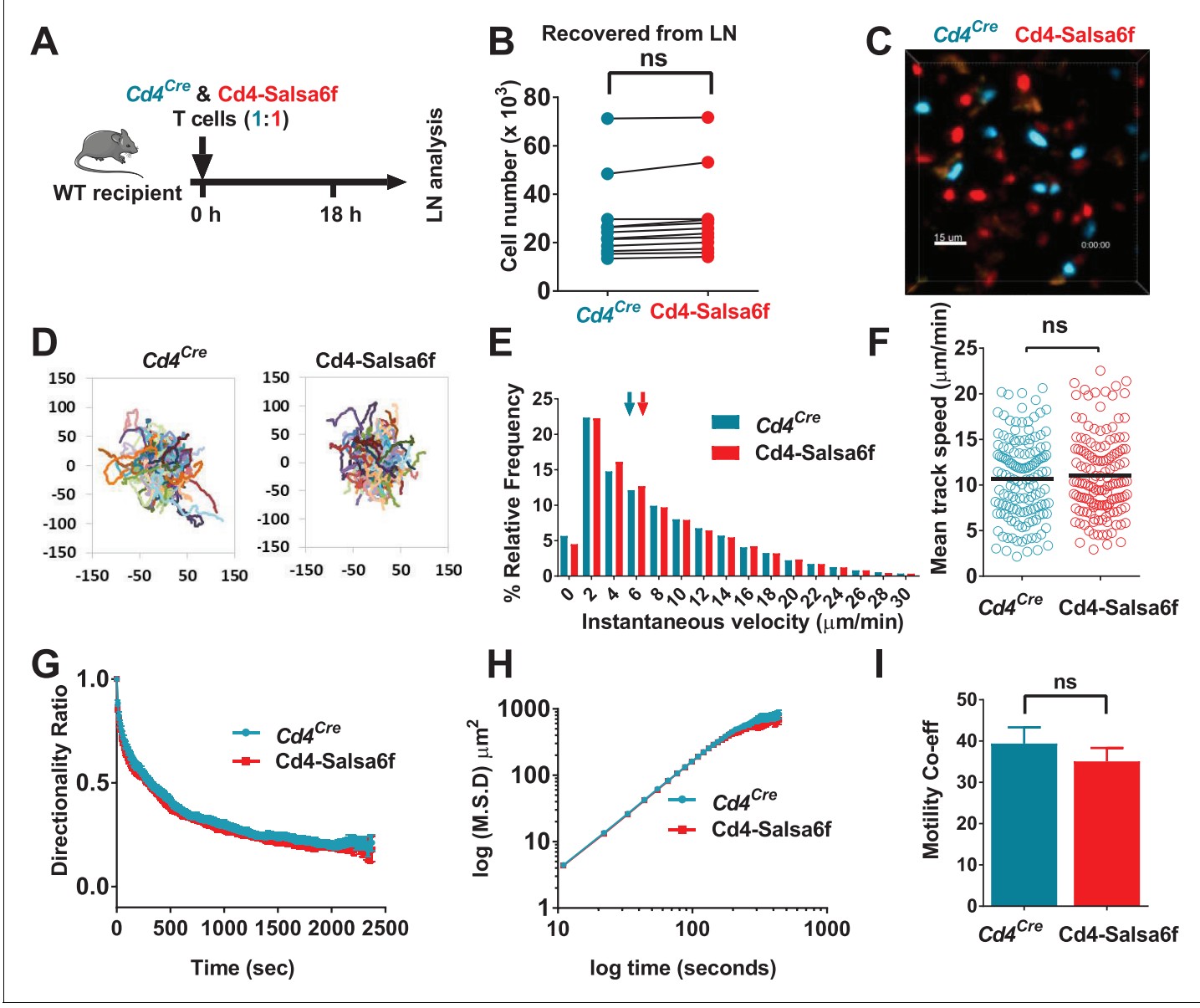

**Figure 6.** Motility of Salsa6f T cells in lymph node following adoptive transfer. *Cd4^Cre* and Cd4-Salsa6f (Hom) cells are shown in teal and in red, respectively. (A) Experimental design to characterize homing and motility of Cd4-Salsa6f cells. CTV-labeled *Cd4^Cre* cells and CTY-labeled Cd4-Salsa6f cells (1:1) were adoptively transferred into wildtype mice, 18 hr prior to LN harvesting. (B) Paired numbers of CTV⁺ and CTY⁺ cells recovered from lymph nodes (p=0.65, Mann Whitney test). (C) Representative median filtered, maximum intensity projection image showing simultaneously imaged *Cd4^Cre* and Cd4-Salsa6f cells the lymph node, scale bar = 30 μm. See *Video 3*. (D) Superimposed tracks with their origins normalized to the starting point. Cells were tracked for more than 20 min. n = 140. (E) Frequency distribution of instantaneous velocities; arrows indicate median, tick marks at the center of every other bin (n > 14,800, three independent experiments). (F) Scatter plot showing mean track speed, black bars indicate overall mean values (11.1 ± 0.4 and 10.7 ± 0.4 μm/min, for *Cd4^Cre* and Cd4-Salsa6f cells respectively, p=0.69; n = 140). (G) Directionality ratio (displacement/distance) over elapsed time (tau = 461 s for *Cd4^Cre* in teal; tau = 474 s for Cd4-Salsa6f in red. n = 217 time points). (H) MSD vs time, plotted on a log-log scale. (I) Measured motility coefficient from 140 tracks (35.1 ± 3.2 vs 39.4 ± 3.9 μm²/min for *Cd4^Cre* and Cd4-Salsa6f cells, p=0.65).
DOI: https://doi.org/10.7554/eLife.27827.012

in T cells (*Vaeth et al., 2017*). These heteromers appear to simply reduce the flow of $Ca^{2+}$ through the Orai1 channel without targeting additional signaling pathways. In the absence of contradictory evidence, we conclude that in T cells Orai1-E106A acts to produce an essentially complete functional knockdown of Orai1-mediated store-operated $Ca^{2+}$ entry.

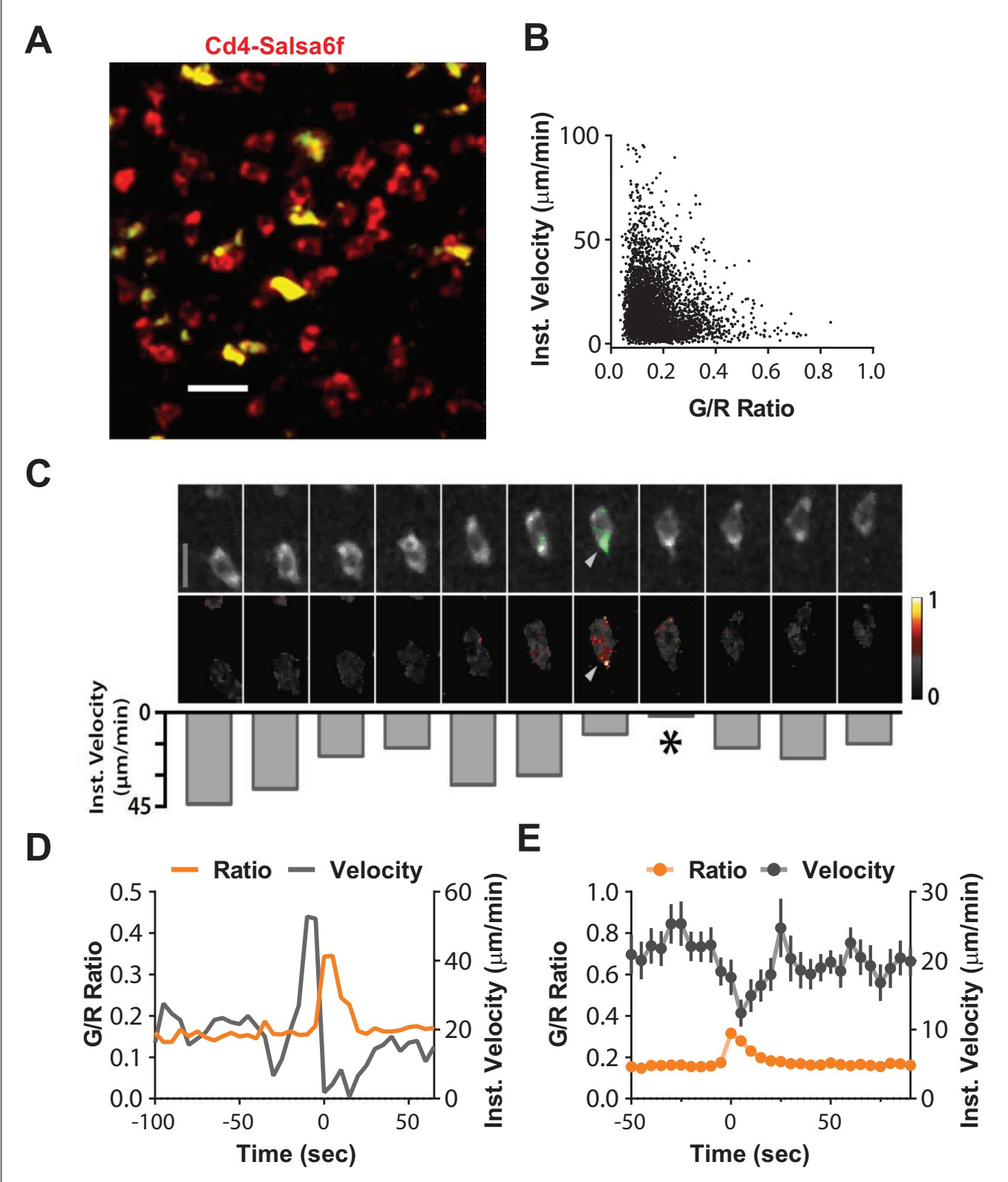

**Figure 7.** Suppression of motility during spontaneous Ca²⁺transients. (A) Median filtered, maximum intensity projection showing cytosolic labeling (exclusion of Salsa6f from the nucleus) in adoptively transferred Cd4-Salsa6f (Hom) cells (red) in the lymph node of wild-type recipients. Autofluorescent structures appear as yellow bodies. Scale bar = 20 µm. See *Video 4*. (B) Scatterplot of instantaneous 3D velocity vs ratio of GCaMP6f (green) to tdTomato (red) fluorescence intensity (r = −0.24, Spearman's rank correlation, p<0.0001, n = 4490 pairs). (C) Image sequence showing a migrating T cell

*Figure 7 continued on next page*

*Figure 7 continued*

and calcium transient from (**A**). Top row: TdTomato signal is shown in grayscale, overlaid with GCaMP6f signals in green. Scale bar = 10 μm. (***Video 5***). Center row: Heat map of Green/Red ratios matched to corresponding images in the top row. Arrows indicate local $Ca^{2+}$ transient. Bottom row: inverted bar graph showing corresponding instantaneous 3D velocities. Asterisk marks a pause in cell motility. (**D**) Representative track from Cd4-Salsa6f (Hom) T cells in lymph nodes, showing intracellular $Ca^{2+}$ levels measured by G/R ratio (orange) on left Y-axis and instantaneous 3D velocity (gray) on right Y-axis. (**E**) Averaged time course of the instantaneous 3D velocity (gray trace, right Y-axis) aligned by the corresponding rise in Salsa6f G/R ratio (orange, left Y-axis). The velocity minimum at time = 5 s-is significantly lower than a baseline from −30 to −10 s (p<0.0001 two-tailed T-test, n = 39 cells).

DOI: https://doi.org/10.7554/eLife.27827.014

Human T cells exhibit systematic changes in motility behavior after Orai1 block, including: increased average velocity, fewer pauses, increased directional persistence and decreased turn angles, and increased motility spread over time. Importantly, maximum and minimum instantaneous velocities are unchanged. For human T cells assessed in motility assays in vitro, similar Orai1-dependent changes are seen. Altered pausing behavior caused by Orai1 block is displayed by isolated single cells inside microchannels formed from photolithographically precise silicon master molds. Changes in motility are dependent upon confinement, as Orai1 block alters pausing under confined but not open-field conditions. The absence of changes to maximum and minimum velocities in vivo and open-field motility in vitro indicates that Orai1 block is not generally deleterious for cell health and movement, but instead acts upon subcellular mechanisms that are selectively employed during cell motility in confined spaces. Taken together, these systematic changes caused by Orai1 block reveal the presence of an Orai1-dependent cell motility program that is utilized frequently enough to be easily detected by changes in the motility characteristics of T cells in the lymph node.

We used Cd4-Salsa6f (Hom) mice to track $Cd4^+$ T cell $Ca^{2+}$ signals in intact mouse lymph nodes and Salsa6f transient transfection to track $Ca^{2+}$ signals in human T cells in vitro and in reconstituted mouse lymph nodes. In our companion paper, *Dong et al., 2017*, we show that Salsa6f expression in $Cd4^+$ T cells is non-perturbing with respect to lymphocyte development, cellular phenotype, cell proliferation, and differentiation. Here, we demonstrate that homing to lymph nodes is unaffected, as are movement patterns within the lymph node, cell velocity, directional persistence, diffusive spread, and motility coefficient. We find that across cells, elevated $Ca^{2+}$ levels are inversely correlated with instantaneous velocity, both in vitro and in vivo. In vivo, moving cells exhibit local $Ca^{2+}$ signals that are strongly associated with pauses in motility. By inspection of movement patterns, turning is likely associated with $Ca^{2+}$ signaling events as well, but this has not been established because most cells move outside of our shallow imaging field either before or after pausing. In many contexts, $Ca^{2+}$ signaling has been shown not only to accompany, but also to cause cell arrest and loss of cell polarity, such as in T cells after activation by antigen (*Negulescu et al., 1996*; *Dustin et al., 1997*; *Wei et al., 2007*). By averaging events, the peak of subcellular $Ca^{2+}$ transients was found to precede the velocity minimum. This event order is consistent with $Ca^{2+}$ causing pauses. While we do not show that the $Ca^{2+}$ signals we observe emanate directly from Orai1 channels, taken together our data are consistent with Orai1 actively regulating cell motility by directly inducing a subcellular motility program that leads to cell arrest.

Two-photon imaging indicated that the frequency of $Cd4^+$ T cells $Ca^{2+}$ transients varies widely between Salsa6f lymph nodes, even when events are normalized for different cell numbers. The origin of this variability is unclear but may result from differences in the distribution and functional properties of APCs within the imaging field. Treatment of MHC class-I and –II blocking antibodies substantially reduces but does not eliminate T cells $Ca^{2+}$ transients. Clearly, a significant number of $Ca^{2+}$ transients are caused by T cell-APC interactions that act through MHC proteins. Given that Orai1 motility events occur frequently as T cell migrate through the lymph node, and $Ca^{2+}$ transients are associated with pauses in motility, we propose that spontaneously generated Orai1-dependent pauses and turns can be triggered by T cell-APC interaction through MHC proteins.

However, we find evidence for MHC-independent triggering of $Ca^{2+}$ signaling and Orai1 channel activation in the lymph node. Human T cells exhibit Orai1-dependent pauses in vitro when migrating as isolated cells in highly uniform microchannels. Salsa6f expression independently detects $Ca^{2+}$ transients in isolated T cells moving within microchannels but not in T cells in adjacent open field portions of the same PDMS imaging chamber. In both cases, responses were produced in the

absence of MHC proteins or APCs. Moreover, we note that, in paired experiments, treatment with MHC class-I and –II blocking antibodies leads to a reduced but notably consistent frequency of $Ca^{2+}$ signaling events. Partial block would be expected to produce substantial variation, especially when combined with a variable input population. Taken together, these data point to the existence of not only MHC-independent Orai1 motility events, but also cell-intrinsic triggering of Orai1. Of note, the apparently random nature of naïve T cell movement in the lymph node has led to the hypothesis that T cells use intrinsic and stochastic motility mechanisms to accomplish immune surveillance (*Wei et al., 2003*; *Mrass et al., 2010*; *Germain et al., 2012*).

In previous studies of Orai1 signaling, Orai1 activation has been placed downstream of extracellular ligand binding to cell surface receptors, integrating their input upon use-dependent depletion of $Ca^{2+}$ from the ER (*Feske, 2007*; *Cahalan and Chandy, 2009*). While we expect signaling downstream of self-antigen detection to be the same as for cognate antigens (*Stefanová et al., 2002*), at this point it is unclear which aspects of internal cell state might lead to cell-intrinsic opening of Orai1 channels and pauses in motility. Of particular interest is determining the step in the signaling cascade from phospholipase C to Orai1 that might be targeted by a novel cell-intrinsic activation pathway. Molecular candidates that underlie regulation of T cell motility by $Ca^{2+}$ are less well defined. One clue to Orai1 action is the subcellular location of $Ca^{2+}$ transients at the back T cells moving within intact lymph nodes, similar to the localization of Orai1 channels during movement in vitro (*Barr et al., 2008*). Early studies demonstrated that immobilization and rounding of T cells bound to antigen-presenting B cells occurred via a calcineurin-independent pathway (*Negulescu et al., 1996*). $Ca^{2+}$-sensitive cytoskeletal proteins, such as myosin II or the actin bundling protein L-plastin, as good candidates for downstream effectors (*Babich and Burkhardt, 2013*; *Morley, 2013*). Like Orai1, Myosin 1 g is selectively required for motility mechanisms under confined conditions (*Gérard et al., 2014*). While Orai1 block reduces pausing but does not otherwise alter T cell velocity, Myo1g block increases pausing and causes cells to move faster. These differences in phenotype suggest that Orai1 and Myo1g act in different, and in part opposing, ways to control T cell motility.

Immune surveillance requires balancing many factors associated with antigen search, including speed and sensitivity (*Friedl and Weigelin, 2008*; *Krummel et al., 2016*). As moving T cells in the lymph node encounter APCs bearing antigen-MHC, they pause due to $Ca^{2+}$-dependent mechanisms unleashed by Orai1 channel opening. These pauses likely ensure adequate time for TCR-antigen scanning by T cell-APC pairs. Our observations of T cell motility indicates that each T cell does not stop at every APC it encounters. Because of this movement pattern, Orai1 provides attractive point of regulation of immune surveillance. Increasing Orai1 activity might be expected to cause T cells to pause more frequently when encountering APCs, restricting the distance T cells move and offering increased opportunities for contact with nearby APCs. Alternatively, decreasing Orai1 activity leads to fewer pauses, greater directional persistence, fewer turns, and greater overall diffusive spread. In this way, Orai1 channel activity could tailor T cell excursions to match the density and reach of dendritic cells in the lymph node. Finally, our findings provide further evidence that during resting conditions, TCR interactions with self-MHC antigens drive continual but limited activation of downstream signaling pathways.

We note some differences between our study and others involving Orai1, STIM1, and T cell motility. These differences might be accounted for, in part, by the expected consequences of our Orai1 dominant-negative approach: block of all three Orai isoforms, limited time for compensatory changes in cell function, and restriction of Orai block to T cells that are adoptively transferred after transfection. (1) Orai1/2 and STIM1/2 KOs have been reported to home to lymph node like wild type, unlike our results here and in a previous paper (*Greenberg et al., 2013*; *Waite et al., 2013*; *Vaeth et al., 2017*). (2) Maximal T cell velocity in the lymph node requires the action of the integrin LFA-1 and the chemokine receptor CCR7 (*Davalos-Misslitz et al., 2007*; *Katakai et al., 2013*), which we have previously shown to be required for entry of T cells into lymph nodes and to act in an Orai1-dependent manner (*Greenberg et al., 2013*). Based upon these findings, blocking Orai1 would be expected to reduce CCR7 and LFA-1 function during interstitial motility as well, resulting in a decrease in T cell velocity. Instead, we find the opposite: Orai1 block leads to an increase in average cell velocity. The absence of any Orai1-dependent change in maximum velocity strongly suggests that CCR7 and LFA-1 do not act through Orai1 during motility in the lymph node. Regardless, any motility effects of LFA-1 and CCR7 are more than compensated by the reduction in pauses caused by Orai1 E106A expression. (3) Previous studies using unconfined open field motility assays

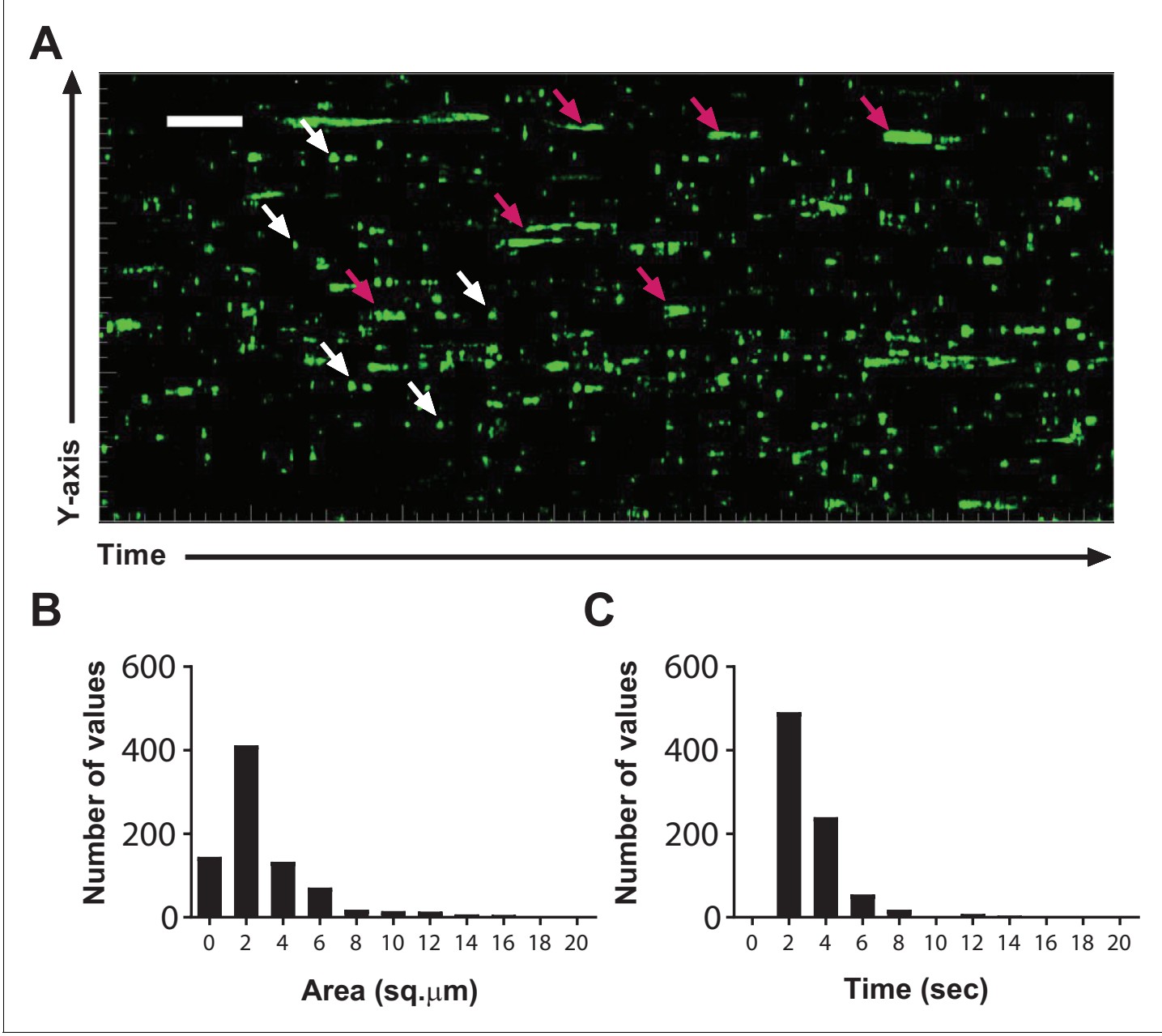

**Figure 8.** T cell Ca²⁺ transients in the steady-state lymph node. (**A**) Calcium history map of steady-state lymph node. Maximum intensity YT projection of 1200 processed green channel time points showing localized sparkles (white arrows) and cell-wide global Ca²⁺ transients (magenta arrows). Scale bar = 50 μm along Y axis, 50 s along T axis. See *Video 6*. (**B**) Frequency distribution of the area of local Ca²⁺ signals. (**C**) Frequency distribution of the duration of local Ca²⁺ signals.

DOI: https://doi.org/10.7554/eLife.27827.017

have excluded a role for Orai1 in T cell motility (*Svensson et al., 2010*; *Kuras et al., 2012*). Our experiments confirm that Orai1 block does not detectably affect unconfined motility; in contrast, our studies in reconstituted lymph nodes and in confined microchannels in vitro both exhibit Orai1-dependent effects. Others have shown that confined motility in vitro better recapitulates mechanisms of motility found in T cells in intact lymph nodes (*Jacobelli et al., 2010*; *Krummel et al., 2016*).

In conclusion, we reveal the existence of an Orai1-dependent cell motility program that leads to pausing of T cells moving within lymph nodes. Imaging with the newly developed genetically

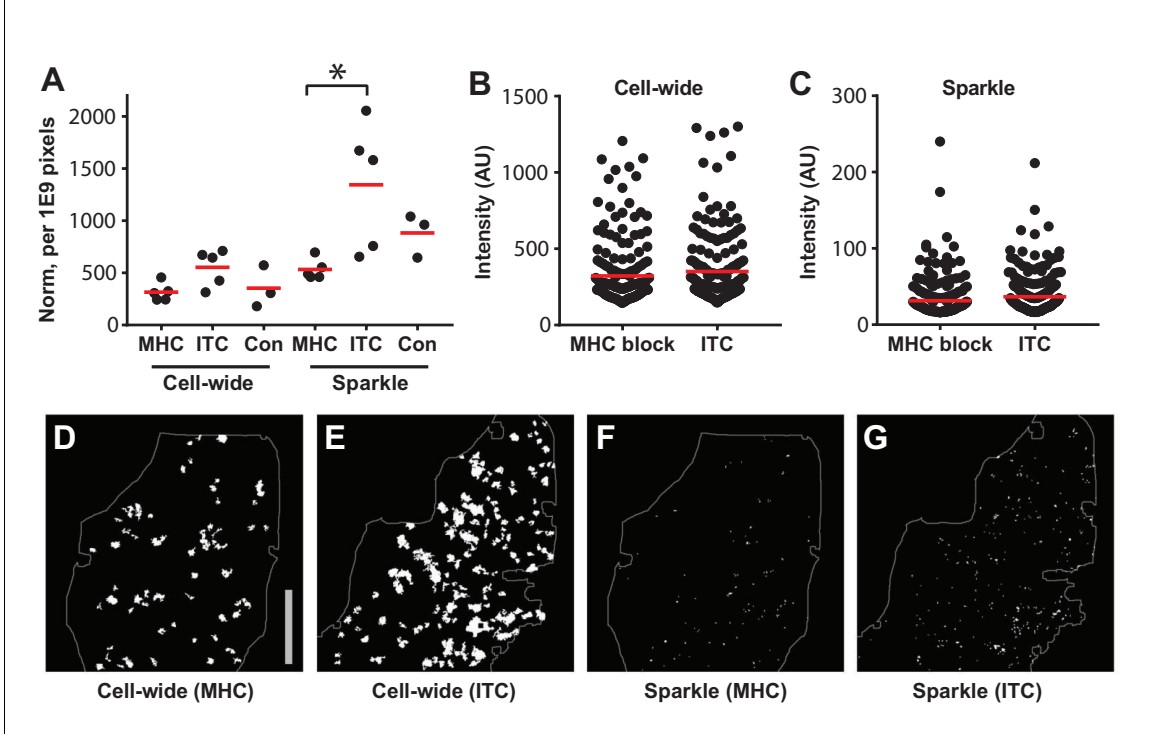

**Figure 9.** MHC block and Ca²⁺transients in steady state lymph nodes. (A) The frequency of cell-wide and local (sparkles) Ca²⁺ transients in CD-Salsa6f (Hom) lymph nodes 48 hr after injection of MHC class I and II blocking antibodies (MHC), isotype control antibody (ITC), or no antibody (Con). Red bars indicate mean values. For MHC-blocked compared to ITC, the relative event frequencies were: for cell-wide, 314 ± 38 vs 553 ± 77, mean ± SEM, p=0.06; for sparkles, 532 ± 44 vs 1343 ± 272, mean ± SEM, p=0.02, Mann-Whitney test. (B,C) Integrated green channel intensities of Ca²⁺ transients normalized to SD of green channel for cell-wide events (B) and for sparkles (C). Red bars indicate mean values. For MHC-blocked vs ITC, the relative amplitudes were: for cell-wide (B), 321 ± 14 vs 350 ± 15, mean ± SEM; for sparkles (C), 32 ± 2 vs 37 ± 2, mean ± SEM. (D–G) Representative thresholded images showing cell-wide and local Ca²⁺ transients, 48 hr after treatment with anti MHC I and II or ITC antibody. The area of the imaging field analyzed is indicated. Scale bar = 100 µm.

DOI: https://doi.org/10.7554/eLife.27827.019

encoded Ca²⁺ indicator Salsa6f identifies local transient Ca²⁺ signaling events with the expected characteristics of Orai1 Ca²⁺ signals. We provide evidence that Orai1-dependent pauses in T cells are triggered in at least two different ways: by self-peptide MHC complexes displayed on the surface of APCs and by a novel cell intrinsic mechanism within the T cells themselves. Together these mechanisms generate motility patterns that promote efficient scanning for cognate antigens in the lymph node.

# Materials and methods

### Key resources table

| Reagent type (species) or resource | Designation | Source or reference | Identifiers | Additional information |
|---|---|---|---|---|
| recombinant DNA reagent | Salsa6f | Dong et al (doi: 10.7554/eLife. 32417) | | |
| strain, strain background (mouse) | Cd4-Salsa6f (Het), Cd4-Salsa6f (Hom) | Dong et al (doi: 10.7554/eLife. 32417) | | |
| strain, strain background (mouse) | NOD.Cg-*Prkdc^scid^B2m^tm1Unc^*/J (NOD.SCID.β2) | Jackson #002570 | | |
| strain, strain background (mouse) | NOD.CB17-*Prkdc^scid^*/J (NOD.SCID) | Jackson #001303 | | |
| strain, strain background (mouse) | *Cd4^Cre^* | Jackson #017336 | | |
| strain, strain background (mouse) | C57BL/6J | Jackson #000664 | | |
| transfected construct (synthetic) | eGFP-Orai1-E106A, eGFP-Orai1 | 23455504 | | |

*Continued on next page*

Continued

| Reagent type (species) or resource | Designation | Source or reference | Identifiers | Additional information |
|---|---|---|---|---|
| biological sample (human) | Primary T cells from healthy human subjects | UCI | | IRB approved |
| antibody | anti-MHC II (Clone Y3P), anti-MHC I (Clone AF6-88.5.5.3), IgG2a Isotype control (Clone: C1.18.4) | BioXCell | | |
| antibody | anti-NK cell antibody | Wako Chemicals | | |
| antibody | αCD3 and αCD28 | Tonbo Biosciences | | |
| peptide, recombinant protein | recombinant human ICAM | R&D Systems | | |
| peptide, recombinant protein | recombinant human IL-2 | BioLegend | | |
| commercial assay or kit | EasySep human T Cell Isolation Kit | Stemcell Technologies | | |
| commercial assay or kit | EasySep mouse T Cell Isolation Kit | Stemcell Technologies | | |
| commercial assay or kit | Nucleofection kit | Lonza | | |
| commercial assay or kit | Sylgard Elastomer 184 kit | Dow Corning | | |
| chemical compound, drug | Cell tracker CMTMR, CellTrace Yellow or CellTrace Violet | Life Technologies | | |
| software, algorithm | Imaris | Bitplane | | |

## Mice and antibodies

NOD.Cg-$Prkdc^{scid}B2m^{tm1Unc}$/J (NOD.SCID.β2) and NOD.CB17-$Prkdc^{scid}$/J (NOD.SCID) mice obtained from Jackson Laboratory (Stock #002570 and #001303) were housed and monitored in a selective pathogen-free environment with sterile food and water in the animal housing facility at the University of California, Irvine. NOD.SCID.β2 mice were reconstituted with human peripheral blood leukocytes (PBLs) as described previously (*Mosier et al., 1988*). A total of $3 \times 10^7$ human PBLs were injected i.p., and experiments were performed 3 weeks later. To inhibit NK cell activity, NOD.SCID mice were i.p. injected with 20 μL anti-NK cell antibody (rabbit anti-Asialo GM1, Wako Chemicals, Irvine, CA) according to manufacturer's instructions 3–4 days before adoptive transfer of human T cells. Mice used were between 8 and 18 weeks of age. The mouse strain expressing Salsa6f selectively in T cells under Cd4-Cre recombinase is described in the comapanion manuscript *Dong et al., 2017*. In brief, LSL-Salsa6f (tdTomato-V5-GCaMP6f) mouse strain was generated in the C57BL/6N background, as described in the accompanying manuscript, and subsequently crossed to homozygotic $Cd4^{Cre}$ mice (JAX #017336) to generate Cd4-Salsa6f (Het) mice expressing Salsa6f only in T cells. These mice were further bred to generate homozygotic Cd4-Salsa6f (Hom) mice for increased Salsa6f expression and fluorescence. Age- and sex-matched C57BL/6J mice from Jackson Laboratory (stock #000664) were used as wild-type recipients of Cd4-Salsa6f (Hom) T cells. To block TCR-MHC interactions, 2 mg of anti-MHC II (Clone Y3P) and 2 mg of anti-MHC I (Clone AF6-88.5.5.3) or 4 mg of IgG2a Isotype control (Clone: C1.18.4) antibodies (Bio X cell) were injected into Cd4-Salsa6f (Hom) litter mates (i.p) 48 hr before imaging.

## Human T cell preparation for imaging

Human PBMCs were isolated from blood of voluntary healthy donors by Histopaque-1077 (1.077 g/mL; Sigma, St. Louis, MO) density-gradient centrifugation, and human T cells isolated using the appropriate EasySep T Cell Isolation Kit (StemCell Technologies). Purified human T cells were rested overnight in complete RPMI, then transfected by nucleofection (Lonza, Walkersville, MD), using the high-viability 'U-014' protocol. Enhanced green fluorescent protein (eGFP)-tagged wild-type Orai1, eGFP-tagged Orai1-E106A mutant, Salsa6f (tdTomato-V5-GCaMP6f construct), or empty vector control were transfected as indicated. Human T cells were used for experiments 3–48 hr after transfection. CMTMR control T cells were prepared by labeling with 10 μM CellTracker CMTMR dye (Invitrogen, Carlsbad, CA) for 10 min at 37°C. For in vivo imaging 10 million human T cells were injected into NOD.SCID.β2 or NOD.SCID mice as indicated. For in vitro imaging experiments, T cells were rested for 3–4 hr in complete RPMI, then washed and activated on plate-bound αCD3 and

αCD28 (Tonbo Biosciences, San Diego, CA) in 2.5 ng/mL recombinant human IL-2 (BioLegend, San Diego, CA), and imaged 24–48 hr after transfection.

## Mouse T cell preparation for imaging

Single cell suspensions of mouse lymphocytes were prepared by mechanical dissociation of spleen and lymph nodes and passing through 70 µm filter. Cd4$^+$ T cells were isolated using the EasySep T Cell Isolation Kit (StemCell Technologies) according to manufacturer's instructions. The purity of isolated cells was confirmed to be >95% by flow cytometry. To compare motility characteristics, Cd4-Salsa6f (Hom) and $Cd4^{Cre}$ control cells were labeled with 10 µM CellTrace Yellow or CellTrace Violet, respectively, for 20 min at 37°C. To measure Ca$^{2+}$ during T cell motility, unlabeled Cd4-Salsa6f (Hom) T cells were adoptively transferred into wild-type recipients. A total of 3–10 million T cells were injected into recipient mice in adoptive transfer experiments (*i.v*: tail-vein or retro-orbital). For confocal imaging on open-field ICAM-1-coated coverslips, Cd4$^+$ T cells from Cd4-Salsa6f (Het) mice were differentiated into Th1 cells using 25 ng/mL rmIL-12 (BioLegend), 10 µg/mL αmouse IL4 (Biolegend) for 4–6 days.

## Microchannel fabrication and imaging

Microchannel fluidic devices were fabricated by a soft lithography technique with PDMS (polydimethylsiloxane; Sylgard Elastomer 184 kit; Dow Corning, Auburn, MI) as described (*Jacobelli et al., 2010*; *Gérard et al., 2014*). PDMS base and curing agent were mixed 10:1 and poured onto the silicon master, then left overnight in vacuum. Once the PDMS was set, it was baked at 55°C for 1 hr and cooled at room temperature. The embedded microchambers were then cut from the mold, and a cell well was punched adjacent to entry into the channels. The PDMS cast and a chambered coverglass (Nunc Lab-Tek, ThermoFisher, Grand Island, NY) were activated for 2 min in a plasma cleaner (Harrick Plasma, Ithaca, NY), bonded together, then incubated at 55°C for 10 min. Prepared chambers were stored for up to 1 month before use. Prior to imaging, microchambers placed in the plasma cleaner for 5 min under vacuum and 1 min of activation, then coated with 5 µg/mL recombinant human ICAM-1/CD54 Fc (R and D Systems, Minneapolis, MN) in PBS for at least 1 hr at 37°C. The microchambers were then washed three times with PBS, and T cells were loaded into cell wells (3−5 × 10$^5$ cells resuspended in 10 µL) and incubated at 37°C for at least 1 hr before imaging.

## Confocal imaging and analysis

Two different Olympus confocal microscopy systems were used to image T cells in vitro. For experiments tracking T cell motility in microchambers, we used the self-contained Olympus Fluoview FV10i-LIV, with a 473 nm diode laser for excitation and a 60x phase contrast water immersion objective (NA 1.2). The FV10i-LIV contains a built-in incubator set to 37°C, together with a Tokai-Hit stagetop incubator to maintain local temperature and humidity. T cells were imaged in RMPI adjusted to 2 mM Ca$^{2+}$ and 2% FCS, and mounted at least half an hour before imaging to allow for equilibration. Cells were imaged at 20 s intervals for 20–30 min, and the data analyzed using Imaris software. For Ca$^{2+}$ imaging of Salsa6f transfected T cells, we used a Fluoview FV3000RS confocal laser scanning microscope, equipped with high-speed resonance scanner and the IX3-ZDC2 Z-drift compensator. Diode lasers (488 and 561 nm) were used for excitation, and two high sensitivity cooled GaAsP PMTs were used for detection of GCaMP6f and tdTomato. Cells were imaged using the Olympus 40x silicone oil objective (NA 1.25), by taking four slice z-stacks at 1.5 µm/step, at 3 s intervals, for up to 20 min. Temperature, humidity, and CO$_2$ were maintained using a Tokai-Hit WSKM-F1 stagetop incubator. Data were processed and analyzed using Imaris software.

## Two-photon imaging and analysis

Multi-dimensional (x, y, z, time, emission wavelength) two-photon microscopy was employed to image fluorescently labeled lymphocytes in explanted mouse lymph nodes, as described (*Miller et al., 2002*; *Matheu et al., 2015*). The following wavelengths were used to excite single or combinations of fluorophores: 900 nm to excite eGFP and CMTMR; 800 nm to excite cell trace violet (Thermofisher C34557) and cell trace yellow (Thermofisher, C34567); 920 nm to excite tdTomato and GCaMP6f; Fluorescence emission was split by 484 nm and 538 nm dichroic mirrors into three detector channels, used to visualize CellTrace Violet or second harmonic signal generated from

collagen in blue; GCaMP6f or eGFP-Orai1E106A transfected cells in green; tdTomato or CellTrace Yellow or CMTMR-labelled cells in red. For imaging, lymph nodes were oriented with the hilum away from the water dipping microscope objective (Olympus 20x, NA 0.9 or Nikon 25x, NA 1.05) on an upright microscope (Olympus BX51). The node was maintained at 36–37°C by perfusion with medium (RPMI) bubbled with carbogen (95% $O_2$/5% $CO_2$). For imaging of human T cells 3D image stacks of x = 200 µm, y = 162 µm, and z = 50 µm were sequentially acquired at 18–20 s intervals using MetaMorph software (Molecular Devices, Sunnyvale, CA). For tracking adoptively transferred mouse T cells, 3D image stacks of x = 250 µm, y = 250 µm, and z = 20 or 52 µm (Voxel size 0.48 µm x 0.48 µm x 4 µm) were sequentially acquired at 5 or 12 s intervals, respectively, using image acquisition software Slidebook (Intelligent Imaging Innovations) as described previously (*Matheu et al., 2015*). This volume collection was repeated for up to 40 min to create a 4D data set. For fast imaging of Cd4-Salsa6f (Hom) lymph nodes, we acquired 2DT images of 300 µm x 300 µm (pixel size 0.65 × 0.65 µm) every 0.5 s. For comparing $Ca^{2+}$ transients in MHC blocking experiments, 3D image stacks of x = 350 µm, y = 350 µm, and z = 20 µm (Voxel size 0.65 µm x 0.65 µm x 4 µm) were sequentially acquired at 5 s intervals. Cell motility data were processed and analyzed using Imaris software (Bitplane USA, Concord, MA). A combination of manual and automatic tracking was used to generate highly accurate cell tracks. The x,y,z coordinates of the tracks were used to calculate speed, M.S.D, directionality ratio, motility coefficients, and to plot tracks as described previously (*Gorelik and Gautreau, 2014*; *Matheu et al., 2015*). Calcium transient (sparkles and cell-wide) analysis and estimation of duration was performed as described previously (*Dong et al., 2017*). XYT data was processed to mask autofluorescent structures, and time was mapped on the Z axis for the purpose of $Ca^{2+}$ transient identification. $Ca^{2+}$ transients were identified in Imaris by a surface-based object identification approach, after manual thresholding of intensity, voxel size (>10) and 2 s minimum duration. Objects were modeled as ellipsoids; X and Y diameter measurements of surfaces were used to calculated areas, and Z diameter (time) was used to estimate duration of $Ca^{2+}$ transients. For MHC-block experiments to estimate the number and intensities of $Ca^{2+}$ transients, we utilized maximum intensity projections from 6 Z stacks. Integrated intensities were normalized to standard deviations of the green channel for comparison of brightness of $Ca^{2+}$ transients.

## Data analysis and statistical testing

Samples sizes were comparable to previous single cell analyses of motility (*Jacobelli et al., 2010*; *Greenberg et al., 2013*; *Gérard et al., 2014*). Each experiment used separate isolations of human T cells from different donors. With the exception of instantaneous velocities in *Figure 6C*, each measurement corresponds to a different cell. Mean ± standard error of the mean was used as a measure of the central tendency of distributions. Video analysis was performed using Imaris software, Spots analysis was used for tracking of cell velocity and Volumes analysis was used for measuring total fluorescence intensity of GECI probes. To reduce selection bias in our analysis of motility and trajectory, all clearly visible and live cells were tracked from each video segment. The arrest coefficient is defined as fraction of time each cell had an instantaneous velocity <2 µm/min. The coefficient of variation was defined for each individual cell as the standard deviation divided by the mean of its instantaneous velocity. For Salsa6f imaging analysis, ratio (R) was calculated by total GCaMP6f intensity divided by total tdTomato intensity, while initial ratio ($R_0$) was calculated by averaging the ratios of the first five time points in each individual cell trace. Photobleaching of tdTomato fluorescence intensity (20–30% decline) was corrected in ratio calculations, as a linear function of time. Figures were generated using Prism 6 (GraphPad Software, San Diego, CA) and Origin 5 (OriginLabs, Northampton, MA). Due to the expectation that individual cells exhibit multiple motility modes, and to avoid assumptions concerning the shapes of motility distributions, non-parametric statistical testing was performed (Mann-Whitney *U* test, unpaired samples, two-tailed, Spearman's rank correlation). Differences with a p value of $\leq 0.05$ were considered significant: *$p \leq 0.05$; **$p < 0.01$; ***$p < 0.005$; ****$p < 0.001$. Similar distributions were compared using the Hodges-Lehmann median difference value and 95% confidence intervals under the assumption that the starting distributions had similar shapes.

## Acknowledgements

We thank Angel Zavala and Drs. Luette Forrest and Olga Safrina for expert assistance, excellent animal care and vivarium support, and Dr. Audrey Gerard, the Matthew Krummel Lab at UCSF, and the Christopher Hughes Lab at UCI for assistance in establishing the microchamber fabrication technique. We acknowledge the UC Irvine Institute for Clinical and Translational Science, and Dr. Jennifer Atwood of the Flow Core Facility supported by the UC Irvine Institute of Immunology.

## Additional information

### Funding

| Funder | Grant reference number | Author |
|---|---|---|
| National Institutes of Health | NS-14609 | Michael D. Cahalan |
| National Science Foundation | IGERT DGE-1144901 | Tobias X. Dong |
| National Institutes of Health | AI-121945 | Michael D. Cahalan |
| National Institutes of Health | GM-41514 | Michael D. Cahalan |
| National Institutes of Health | Training Grant T32-AI-060573 | Milton L. Greenberg |
| National Institutes of Health | Training Grant T32-GM-008620 | Tobias X. Dong |

The funders had no role in study design, data collection and interpretation, or the decision to submit the work for publication.

### Author contributions

Tobias X Dong, Conceptualization, Data curation, Formal analysis, Funding acquisition, Investigation, Visualization, Methodology, Writing—original draft, Writing—review and editing; Shivashankar Othy, Data curation, Formal analysis, Validation, Investigation, Visualization, Methodology, Writing—review and editing; Milton L Greenberg, Conceptualization, Data curation, Formal analysis, Methodology, Writing—original draft, Writing—review and editing; Amit Jairaman, Conceptualization, Data curation, Formal analysis, Investigation, Methodology, Writing—original draft, Writing—review and editing; Chijioke Akunwafo, Data curation, Formal analysis, Methodology; Sabrina Leverrier, Data curation, Formal analysis, Investigation, Methodology, Writing—review and editing; Ying Yu, Conceptualization, Resources, Data curation, Formal analysis, Supervision, Investigation, Visualization, Methodology, Writing—original draft, Writing—review and editing; Ian Parker, Conceptualization, Resources, Data curation, Formal analysis, Supervision, Visualization, Methodology, Writing—original draft, Writing—review and editing; Joseph L Dynes, Conceptualization, Data curation, Formal analysis, Visualization, Methodology, Writing—original draft, Writing—review and editing; Michael D Cahalan, Conceptualization, Resources, Supervision, Funding acquisition, Investigation, Methodology, Writing—original draft, Project administration, Writing—review and editing

### Author ORCIDs

Tobias X Dong (iD) https://orcid.org/0000-0001-5500-7099
Shivashankar Othy (iD) http://orcid.org/0000-0001-6832-5547
Michael D Cahalan (iD) http://orcid.org/0000-0002-4987-2526

### Ethics

Human subjects: All studies using human blood were approved by the University of California, Irvine Institutional Review Board (UCI IRB HS# 1995-460), and complied with all applicable UCI Research Policies for the conduct of human subjects research.
Animal experimentation: All surgical procedures and animal maintenance complied with the National Institute of Health guidelines regarding the care and use of experimental animals and were approved by the Animal Care and Use Committee of University of California, Irvine (protocol 1998-1366).

### Decision letter and Author response

Decision letter https://doi.org/10.7554/eLife.27827.022
Author response https://doi.org/10.7554/eLife.27827.023

---

## Additional files

### Supplementary files

• Transparent reporting form
DOI: https://doi.org/10.7554/eLife.27827.020

---

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
