## [Decision Letter]

Thank you for submitting your article "Cell-intrinsic activation of Orai1 regulates human T cell motility" for consideration by *eLife*. Your article has been favorably evaluated by Michel Nussenzweig (Senior Editor) and three reviewers, one of whom, Michael L Dustin (Reviewer #1), is a member of our Board of Reviewing Editors.

The reviewers and editors feel that your paper (Cahalan et al.) potentially complements an independently submitted paper (“Calcium-mediated shaping of naive CD4 T-cell phenotype and function” by Auffray et al.). You were contacted about this opportunity and agreed to exchange information with the authors of that paper after receiving the three individual reviews for each. The following is a consensus review among three of the reviewers who agreed to oversee a co-revision. The intention is to help you prioritise the revision and also to clarify the opportunity for coordination from our perspective.

The two papers deal with different consequences of calcium mobilization in vivo. Cahalan et al. focuses on transcriptional changes that are brought about in the most self-reactive naïve T cells in mice that lead to loss of Ly6C expression and a tendency to differentiate into peripheral Treg upon activation. This state could be mimicked by treatments that may chronically enhance cytoplasmic calcium, but the evidence for any effects on cytoplasmic calcium ion levels by self pMHC in vivo or low concentrations of thapsigargin in vitro are lacking, which is a major gap. There are also mechanistic aspects related to cytokine production in the Ly6C positive population that could be explored.

Cahalan et al. also focuses on the role of CRAC channel activation in T cell trafficking and interstitial migration in the T cell zone of lymph nodes. A unique aspect of the paper is that they focus entirely on human T cells in profoundly immunodeficient mice. CRAC is eliminated through expression of a dominant negative Orai, which is well established to eliminate sustained calcium increases. The authors developed a new calcium reporter based on fusing GCaMP6f to tdTomato to make a ratiometric reporter called Salsa6f, which is sufficient sensitive to detect calcium spikes in intact lymph nodes. The main points are that CRAC channel function is needed for efficient lymph node entry and that cell-autonomous calcium fluxes result in slowing of T cell migration. The weakness in this paper is that the actual mechanism of the calcium fluxes is not established and objective statistical tests supporting a relationship are not provided. The trigger for the calcium fluxes observed is not clear.

The editors felt that there is strong possibility that addressing issues in Cahalan et al. may provide key supporting data for Auffray.et al. And reciprocally, data in Auffray et al. provides further physiological significance for data in Cahalan.et al. The authors may therefore benefit from coordinating on the revisions and the authors have agreed to exchange information. The editors have prepared the following joint consensus review following provision of the individual reviews.

1) The main concern with Auffray et al. is that there is no data on the impact of pharmacological manipulations on cytoplasmic calcium levels in the naïve mouse T cells. The authors of Cahalan et al. are experts in this area. It will be important for the authors to use in vitro calcium studies to determine the impact of chronic exposure to 4 nM thapsigargin in different time windows – does this result in sustained cytoplasmic calcium elevation capable to inducing NFAT translocation? This may require a highly sensitive method and advice from authors of Cahalan et al. should be helpful to address this concern in a 2-month time frame. But this concern needs to be addressed.

2) The main concern with Cahalan et al. is that the mechanism triggering the calcium fluxes in vitro and in vivo is not clear. T cell migrating on flat substrates didn't display the calcium fluxes, but movement in PDMS channels and within lymph nodes did. The authors’ attention was called to recent studies that suggest that surface interactions could trigger calcium signals related to formation of close contacts that exclude CD45. Thus, some cellular confinement in the PDMS channels may trigger signals. The in vivo situation is of greatest interest. The use of the human-mouse system is a complication as its not entirely clear self pMHC is supplied by human monocyte derived DC or mouse xeno-MHC, which can trigger T cell responses. In murine systems, particularly on H2-b haplotype, it is relatively easy to eliminate self pMHC signals in CD4^+^ T cells acutely by blocking I-Ab with an antibody (e.g. Stefanova I, Dorfman JR, Germain RN. Self-recognition promotes the foreign antigen sensitivity of naive T lymphocytes. Nature. 2002;420(6914):429-34) or by transferring CD4 T cells in to I-Ab deficient mice. Thus, while the human mouse system has some utility as a preclinical model, it's a relatively weak tool to study basic mechanisms in this case as even if only human pMHC is recognized, there are multiple possible class I and class II genes that would need to be blocked to eliminate all possible recognition. The author should repeat a key experiment in a mouse model system where a role of self-pMHC in the in vivo calcium fluxes could be supported or ruled out. Introduction of the Salsa6f reporter into naïve T cells would be helpful in this context. If the authors happen to have generated a transgenic model with Salsa6f or have the ability to introduce it by RNA electroporation then this could be combined with antibody blocking to test a role for self-pMHC in the calcium activity independent of the DN Orai construct, which is not necessarily needed to ask this question. The result of this experiment would also provide key information for Auffray.et al.

Additional points:

1) Additional statistical analysis of the migration in lymph nodes should be undertaken to better understand the statistical relationship between calcium fluxes and T cell turning and pausing behavior. Is there a statistical correlation that can be identified?

2) Can the authors perform in in vivo calibration such that calcium levels in the DN Orai and control T cells could be directly compared? Is there a difference in the average calcium levels, beyond the spiking?

---

## [Author Response]

[…] 1) The main concern with Auffray et al. is that there is no data on the impact of pharmacological manipulations on cytoplasmic calcium levels in the naïve mouse T cells. The authors of Cahalan et al. are experts in this area. It will be important for the authors to use in vitro calcium studies to determine the impact of chronic exposure to 4 nM thapsigargin in different time windows – does this result in sustained cytoplasmic calcium elevation capable to inducing NFAT translocation? This may require a highly sensitive method and advice from authors of Cahalan et al. should be helpful to address this concern in a 2-month time frame. But this concern needs to be addressed.

We provided advice on the experiments to Dr. Auffray and advised caution about the potential hazards of using pharmacological manipulations due to lack of potency and specificity of CRAC channel blockers. We advised to expand the calcium measurements to include different time‐points and not to use a “delta measurement”, instead to focus on the baseline levels. We also cautioned that upregulation of PMCA activity could explain some of the results. In addition, we consulted on the conditions for pMHC antibody block and are assured that our experiments with the same antibody are compatible.

2) The main concern with Cahalan et al. is that the mechanism triggering the calcium fluxes in vitro and in vivo is not clear. T cell migrating on flat substrates didn't display the calcium fluxes, but movement in PDMS channels and within lymph nodes did. The authors’ attention was called to recent studies that suggest that surface interactions could trigger calcium signals related to formation of close contacts that exclude CD45. Thus, some cellular confinement in the PDMS channels may trigger signals. The in vivo situation is of greatest interest. The use of the human-mouse system is a complication as its not entirely clear self pMHC is supplied by human monocyte derived DC or mouse xeno-MHC, which can trigger T cell responses. In murine systems, particularly on H2-b haplotype, it is relatively easy to eliminate self pMHC signals in CD4^+^ T cells acutely by blocking I-Ab with an antibody (e.g. Stefanova I, Dorfman JR, Germain RN. Self-recognition promotes the foreign antigen sensitivity of naive T lymphocytes. Nature. 2002;420(6914):429-34) or by transferring CD4 T cells in to I-Ab deficient mice. Thus, while the human mouse system has some utility as a preclinical model, it's a relatively weak tool to study basic mechanisms in this case as even if only human pMHC is recognized, there are multiple possible class I and class II genes that would need to be blocked to eliminate all possible recognition. The author should repeat a key experiment in a mouse model system where a role of self-pMHC in the in vivo calcium fluxes could be supported or ruled out. Introduction of the Salsa6f reporter into naïve T cells would be helpful in this context. If the authors happen to have generated a transgenic model with Salsa6f or have the ability to introduce it by RNA electroporation then this could be combined with antibody blocking to test a role for self-pMHC in the calcium activity independent of the DN Orai construct, which is not necessarily needed to ask this question. The result of this experiment would also provide key information for Auffray et al.

We thank the reviewer for the prescient suggestion of using a calcium‐reporter transgenic mouse to extend our approach and to address questions related to pMHC. Indeed, relevant to the main concern and reviewer comment highlighted in bold above, we have made a Cre‐dependent transgenic mouse to allow subset‐specific expression of Salsa6f driven by CD4‐ and other Cre‐dependent driver lines, and we submitted a second manuscript to *eLife* for consideration as the first of two back‐to‐back publications. The proposed companion paper is titled: “T Cell Calcium Dynamics Visualized in a Ratiometric tdTomato‐GCaMP6f Transgenic Reporter Mouse”. It presents the breakthrough capabilities of the transgenic Salsa 6f mouse presents for imaging calcium signaling in a cell‐specific manner in isolated cells and in the complex tissue environment in vivo. Our proposed companion manuscript introduces the new probe, Salsa6f, and demonstrates advantages for Ca^2+^ monitoring in T cells from CD4‐Cre‐Salsa6f transgenic mice, including the following outline of the Results section:

• A novel ratiometric genetically encoded Ca^2+^ indicator, Salsa6f;

• Generation of Salsa6f transgenic reporter mice and validation in immune cells;

• Single‐cell ratiometric Ca^2+^ measurement in CD4‐Cre‐Salsa6f reporter mice;

• Cytosolic localization and calibration of Salsa6f in transgenic T lymphocytes;

• T cell Ca^2+^ signaling in response to Ca^2+^ store depletion, T cell receptor engagement, and mechanical stimulation;

• TCR‐induced Ca^2+^ signaling in naïve and differentiated helper T cell subsets;

• Two‐photon microscopy of CD4Cre^+/‐^Salsa6f^+/‐^ T cells in lymph node revealing novel spontaneous Ca^2+^ signaling during basal motility in the absence of specific antigen.

Of particular note and related to our revision of this manuscript, we were able to detect spontaneous, transient Ca^2+^ signals in T cells migrating in lymph node under steady‐state conditions. Our revised manuscript would then follow our companion paper. We now include a section fully describing Ca^2+^ signals detected using Salsa6f in transgenic mouse T cells during basal motility under homeostatic conditions in mouse lymph node and in the absence of specific antigen. We call these localized signals sparkles because of their appearance on rapid playback; and show that they are correlated with pausing and turning behavior. They are inhibited by antibody treatment to peptide MHC Class I and II. We suggest that putting the two papers back to back will not only strengthen our manuscript and address reviewer comments, but will attract a great deal of interest in the calcium signaling field, which would include immunologists as well as neuroscientists.

We would be happy to consider if the new data set include experiments that address the role of selfpMHC recognition, to rule it in or out. This is key to the potential synergy with the Auffray paper and we feel its intrinsic to any demonstration that the calcium signals don't represent "antigen" as the cells are differentially sensing self and foreign pMHC. So if "antigen" includes explicit data on self this should be great. One concept that might be interesting to consider is that a truly antigen (self or foreign) independent Calcium signal could generate white noise that has some impact on motility, but also allows the T cells to detect the sub‐threshold self‐antigen signal and respond to it with calcineurin/NFAT dependent transcription. This might be an example of stochastic resonance and is described in other sensing systems that need to detect weak signals. We agree that testing the role of self and foreign antigen signalling in the new mouse model makes much more sense than any such manipulation in the human‐mouse system described in this paper, which could be revised to address the mechanism of antigen independent Calcium signalling and co‐submitted with the new manuscript describing the new mouse model.

There are four new figures in the motility manuscript (Figure 6‐9), including the peptide‐MHC blocking studies (Figure 9). We found that treatment of MHC class‐I and –II blocking antibodies substantially reduces but does not eliminate T cells Ca^2+^ transients. We hope you and the reviewers will find the new results of interest.

Additional points:1) Additional statistical analysis of the migration in lymph nodes should be undertaken to better understand the statistical relationship between calcium fluxes and T cell turning and pausing behavior. Is there a statistical correlation that can be identified?

We have performed additional analyses of motility characteristics in both human and mouse T cells, including analysis of turning and pausing behavior (Figure 1 and Figure 6). The revised manuscript also includes four new figures on the mouse data describing the relationship of calcium to cellular motility.

2) Can the authors perform in in vivo calibration such that calcium levels in the DN Orai and control T cells could be directly compared? Is there a difference in the average calcium levels, beyond the spiking?

Cytosolic Ca^2+^ calibration of Salsa6f is described in Figure 6‐F of the companion Salsa6f transgenic mouse manuscript (Cahalan et al, companion paper). In this manuscript, we use the green/red ratio to track and compare calcium levels in the mouse experiments.

We appreciate the comments from reviewers, which prompted a major revision with four new figures. The manuscript has been substantially revised and, we hope the reviewers will agree, greatly improved by including new results from CD4‐Cre‐Salsa6f transgenic mice. The description of our new Ca^2+^ indicator, Salsa6f, has been moved to the companion manuscript, titled: “T Cell Calcium Dynamics Visualized in a Ratiometric tdTomato‐GCaMP6f Transgenic Reporter Mouse”. The revised manuscript now includes two‐photon imaging data of CD4‐Salsaf^+/+^ T cells to allow Ca^2+^ to be related to basal cellular motility in the steady‐state lymph node, including:

• Figure 6. Motility of Salsa6f T cells in lymph node following adoptive transfer. [Absolutely normal, motility not affected by the probe, see also Video 3]

• Figure 7. Suppression of motility during spontaneous Ca^2+^ transients. [Directly relates spontaneous Ca^2+^ transients to pauses in motility, Video 4,Video 5]

• Figure 8 cell Ca^2+^ transients in the steady‐state lymph node. [Two types of spontaneous Ca^2+^ events in T cells during basal motility – subcellular sparkles and global transients, Video 6]

• Figure 9. MHC block and Ca^2+^ transients in steady state lymph nodes. [Evaluated by antibodyblock of pMHC class I and II − significant inhibition of sparkle frequency, no change in amplitude, no significant change in cell‐wide Ca^2+^ transients]

In addition, the Discussion was substantially revised to interpret the new results in terms of what is known about T cell motility and antigen scanning in mouse lymph node.